# Provable Robustness against Wasserstein Distribution Shifts via Input Randomization

**Aounon Kumar**
University of Maryland
College Park, MD
`aounon@umd.edu`

**Alexander Levine**
University of Maryland
College Park, MD
`alevine0@cs.umd.edu`

**Tom Goldstein**
University of Maryland
College Park, MD
`tomg@cs.umd.edu`

**Soheil Feizi**
University of Maryland
College Park, MD
`sfeizi@cs.umd.edu`

## Abstract

Certified robustness in machine learning has primarily focused on adversarial perturbations with a fixed attack budget for each sample in the input distribution. In this work, we present provable robustness guarantees on the accuracy of a model under bounded Wasserstein shifts of the data distribution. We show that a simple procedure that randomizes the input of the model within a transformation space is provably robust to distributional shifts under that transformation. Our framework allows the datum-specific perturbation size to vary across different points in the input distribution and is general enough to include fixed-sized perturbations as well. Our certificates produce guaranteed lower bounds on the performance of the model for any shift (natural or adversarial) of the input distribution within a Wasserstein ball around the original distribution. We apply our technique to certify robustness against natural (non-adversarial) transformations of images such as color shifts, hue shifts, and changes in brightness and saturation. We obtain strong performance guarantees for the robust model under clearly visible shifts in the input images. Our experiments establish the non-vacuousness of our certificates by showing that the certified lower bound on a robust model's accuracy is higher than the empirical accuracy of an undefended model under a distribution shift. We also show provable distributional robustness against adversarial attacks. Moreover, our results also imply guaranteed lower bounds (hardness result) on the performance of models trained on so-called "unlearnable" datasets that have been poisoned to interfere with model training. We show that the performance of a robust model is guaranteed to remain above a certain threshold on the test distribution even when the base model is trained on the poisoned dataset.

## 1 Introduction

Machine learning models often suffer significant performance loss under minor shifts in the data distribution that do not affect a human's ability to perform the same task– e.g., input noise (Dodge & Karam, 2016; Geirhos et al., 2018), image scaling, shifting and translation (Azulay & Weiss, 2019), spatial (Engstrom et al., 2019) and geometric transformations (Fawzi & Frossard, 2015; Alcorn et al., 2019), blurring (Vasiljevic et al., 2016; Zhou et al., 2017), acoustic corruptions (Pearce & Hirsch, 2000) and adversarial perturbations (Szegedy et al., 2014; Carlini & Wagner, 2017; Goodfellow et al., 2015; Madry et al., 2018; Biggio et al., 2013). Overcoming such robustness challenges is a major hurdle for deploying these models in safety-critical applications where reliability is paramount. Several techniques have been developed to improve the empirical robustness of a model to data shifts, e.g., diversifying datasets (Taori et al., 2020), training with natural corruptions (Hendrycks & Dietterich, 2019), data augmentations (Yang et al., 2019), contrastive learning (Kim et al., 2020; Radford et al., 2021; Ge et al., 2021) and adversarial training (Goodfellow et al., 2015; Madry et al., 2018; Tramèr & Boneh, 2019; Shafahi et al., 2019; Maini et al., 2020). Empirical

robustness techniques are designed to protect a model against a particular type of shift or adversary (e.g., by introducing similar shifts during training) and may not be effective against new ones. For instance, adversarial defenses have been shown to break down under newer attacks (Carlini & Wagner, 2017; Athalye et al., 2018; Uesato et al., 2018; Laidlaw & Feizi, 2019; Laidlaw et al., 2021).

Certifiable robustness methods, on the other hand, seek to produce provable guarantees on the robustness of a model which hold for any perturbation within a certain neighborhood of the input instance regardless of the strategy used to generate this perturbation. A robustness certificate produces a verifiable lower bound on the size of the perturbation required to fool a model. Apart from being a guarantee on the robust performance, these certificates may also serve as a metric to compare the robustness of different models that is independent of the mechanism producing the input perturbations. However, the study of provable robustness has mostly focused on perturbations with a fixed size budget (e.g., an $\ell_p$-ball of same size) for all input points (Cohen et al., 2019; Lécuyer et al., 2019; Li et al., 2019; Salman et al., 2019; Gowal et al., 2018; Huang et al., 2019; Wong & Kolter, 2018; Raghunathan et al., 2018; Singla & Feizi, 2019; 2020; Levine & Feizi, 2021; 2020a;b). Among provable robustness methods, randomized smoothing based procedures have been able to successfully scale up to high-dimensional problems (Cohen et al., 2019; Lécuyer et al., 2019; Li et al., 2019; Salman et al., 2019) and adapted effectively to other domains such as reinforcement learning (Kumar et al., 2021; Wu et al., 2021) and models with structured outputs (Kumar & Goldstein, 2021) as in segmentation tasks and generative modeling. However, these techniques cannot be extended to certify under distribution shifts as the perturbation size for each instance in the input distribution need not have a fixed bound. For example, stochastic changes in the input images of a vision model caused by lighting and weather conditions may vary across time and location. Even adversarial attacks may choose to adjust the perturbation size depending on the input instance.

A standard way of describing a distribution shift is to constrain the Wasserstein distance between the original distribution $\mathcal{D}$ and the shifted distribution $\tilde{\mathcal{D}}$ to be bounded by a certain amount $\epsilon$, i.e., $W_1^d(\mathcal{D}, \tilde{\mathcal{D}}) \leq \epsilon$, for an appropriate distance function $d$. The Wasserstein distance is the minimum expectation of the distance function $d$ over all possible joint distributions with marginals $\mathcal{D}$ and $\tilde{\mathcal{D}}$. Wasserstein distance is a standard similarity measure for probability distributions and has been extensively used to study distribution shifts (Courty et al., 2017; Damodaran et al., 2018; Lee & Raginsky, 2018; Wu et al., 2019). Certifiable robustness against Wasserstein shifts is an interesting problem to study in its own right and a useful tool to have in the arsenal of provable robustness techniques in machine learning.

In this work, we design robustness certificates for distribution shifts bounded by a Wasserstein distance of $\epsilon$. We show that by simply randomizing the input in a transformation space, it is possible to bound the difference between the accuracy of the robust model under the original distribution $\mathcal{D}$ and the shifted distribution $\tilde{\mathcal{D}}$ as a function of their Wasserstein distance $\epsilon$ under that transformation. Given a base model $\mu$, we define a robust model $\bar{\mu}$ which replaces the input of $\mu$ with a randomized version sampled from a "smoothing" distribution around the original input. Let $\bar{h}$ be a function denoting the performance of the robust model $\bar{\mu}$ on an input-output pair $(x, y)$ (see Section 3 for a formal definition). Then, our main theoretical result in Theorem 1 shows that

$$\left| \mathbb{E}_{(x_1,y_1)\sim\mathcal{D}}[\bar{h}(x_1, y_1)] - \mathbb{E}_{(x_2,y_2)\sim\tilde{\mathcal{D}}}[\bar{h}(x_2, y_2)] \right| \leq \psi(\epsilon),$$

where $\psi$ is a concave function that bounds the total variation between the smoothing distributions at two input points as a function of the distance between them (condition (3) in Section 3). Such an upper bound always exists for any smoothing distribution as the total variation remains between zero and one as the distance between the two distributions increases. We discuss how to find the appropriate $\psi$ for different smoothing distributions in Appendix G.

We apply our result to certify model performance for families of parameterized distribution shifts which include shifts in the RBG color balance of an image, the hue/saturation balance, the brightness/contrast, and more. Our method does not make any assumptions on the model and applies to both natural and adversarial shifts of the distribution. It does not increase the computational requirements of the base model as it only samples one randomized input per robust prediction, making it scalable to high-dimensional problems that require conventional deep neural network architectures. The sample complexity for generating the Wasserstein certificates over the entire distribution is roughly the same as obtaining adversarial certificates for a single input instance using existing randomized smoothing based techniques (Cohen et al., 2019; Salman et al., 2019).

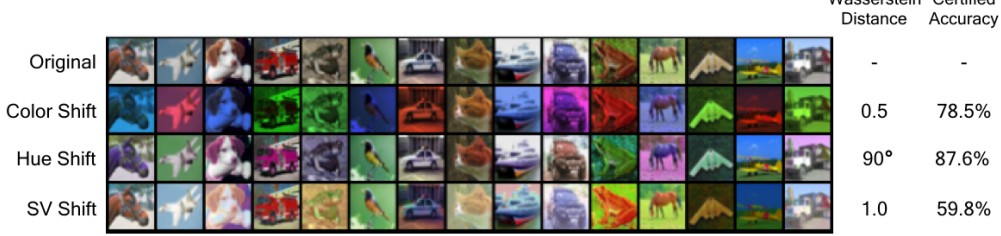

Figure 1: Certified accuracies obtained for different natural transformations of CIFAR-10 images such as color shifts, hue shifts and changes in brightness and saturation. The Wasserstein distance of each distribution shift from the original distribution is defined with respect to the corresponding distance function.

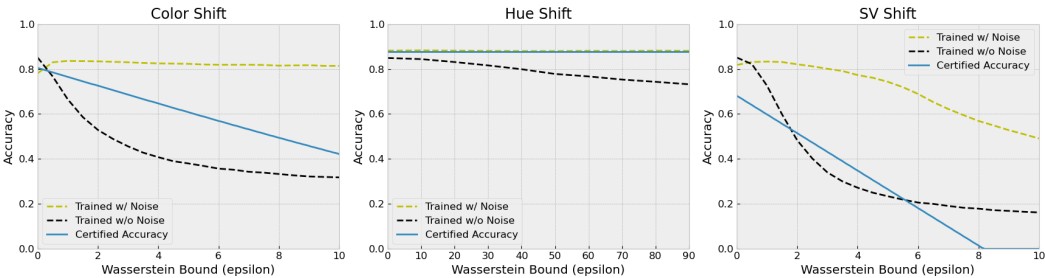

Figure 2: Comparison between the empirical performance (dashed lines) of two base models (trained on CIFAR-10 images with and without noise in transformation space) and the certified accuracy (solid line) of a robust model (noise-trained model smoothed using input randomization) under distribution shifts. The certified accuracy often outperforms the undefended model and remains reasonably close (almost overlaps for hue shift) to the model trained under noise for small shifts in the distribution.

Robustness under distribution shifts is a fundamental problem in several areas of machine learning and our certificates could be applicable to a multitude of learning tasks. We demonstrate the usefulness of our main theoretical result (Theorem 1) in the following domains:

**(i) Certifying model accuracy under natural shifts (Section 5):** We consider three image transformations: color shift, hue shift and changes in brightness and saturation (SV shift). Figure (1) visualizes CIFAR-10 (Krizhevsky et al.) images under each of these transformations and reports the corresponding certified accuracies obtained by our method. Figure (2) plots the accuracy of two base models (trained on CIFAR-10 images with and without noise in the transformation space) under a shifted distribution and compares it with the certified accuracy of a robust model (noise-trained model smoothed using input randomization). These results demonstrate that our certificates are significant and non-vacuous (see appendix I for more details). In figures (3) and (4), we plot the certified accuracies for different values of training and smoothing noise – first for the CIFAR-10 dataset and then confirm our results on the SVHN dataset (Netzer et al., 2011).

**(ii) Certifying population level robustness against adversarial attacks (Section 6):** The distribution of instances generated by an adversarial attack can also be viewed as a shift in the input distribution within a Wasserstein bound. Unlike existing certification techniques which assume a fixed perturbation budget across all inputs (Cohen et al., 2019; Lécuyer et al., 2019; Li et al., 2019; Salman et al., 2019), our guarantees work for a more general threat model where the adversary is allowed to choose the perturbation size for each input instance as long as it respects the constraint on the average perturbation size over the entire data distribution. Also, our procedure only requires *one* sample from the smoothing distribution per input instance which makes computing population level certificates significantly more efficient than existing techniques. The certified accuracy we obtain significantly outperforms the base model under attack (figure 7a).

**(iii) Hardness results for generating "unlearnable" datasets (Section 7):** Huang et al. (2021) proposed a method to make regular datasets unusable for modern deep learning models by poisoning them with adversarial perturbations to interfere with the training of the model. The intended purpose is to increase privacy for sensitive data such as personal images uploaded to social media sites. The dataset is poisoned in such a way that a model that minimizes the loss on this data distribution will have low accuracy on clean test samples. We show that our framework can obtain verifiable

lower bounds on the performance of a model trained on such unlearnable datasets. Our certificates guarantee that the performance of the robust model (using input randomization) will remain above a certain threshold on the test distribution even when the base model is trained on the poisoned dataset with a smoothing noise of suitable magnitude. This demonstrates a fundamental limitation in producing unlearnable datasets.

## 2 RELATED WORK

Several methods for introducing corruptions during training have been shown to improve the empirical robustness of machine learning models (Hendrycks & Dietterich, 2019; Yang et al., 2019; Goodfellow et al., 2015; Madry et al., 2018). Training with input transformations, such as blurring, cropping and rotations, can improve test accuracy against these corruptions. However, these methods do not produce any guarantees on the performance of the model with respect to the amount of shift added to the distribution. Our method applies random input transformations during inference to make the model provably robust against any distribution shift within a certain Wasserstein distance. It is independent of the model architecture and training procedure, and can be coupled with robust training techniques, such as noise or adversarial training, to improve the certified performance.

Randomized smoothing based approaches that aggregate model predictions over a large number of noised samples of the input (Cohen et al., 2019; Lécuyer et al., 2019; Li et al., 2019; Salman et al., 2019) and that use input randomization (Pinot et al., 2021) have been studied in the context of certified adversarial robustness. Provable robustness for parameterized transformations on images also exist (Fischer et al., 2020). These techniques produce instance-wise fixed-budget certificates and do not generate robustness guarantees over the entire data distribution or allow varying perturbation sizes for different instances. Our work also differs from instance-wise adversarial attacks and defenses (Wong et al., 2019; Levine & Feizi, 2019) that use the Wasserstein distance (instead of conventional $\ell_p$ distances) to measure difference between an image and its perturbed version. In contrast, our certificates consider the Wasserstein distance between data distributions from which images themselves are sampled.

Robustness bounds on the population loss against Wasserstein shifts under the $\ell_2$-distance (Shen et al., 2018; Sinha et al., 2018) have been derived assuming Lipschitz-continuity of the base model. These bounds depend on the Lipschitz constant for the underlying model, which can grow rapidly for deep neural networks. We produce guarantees on the accuracy of an arbitrary model without requiring any restrictive assumptions or a global Lipschitz bound. Additionally, our approach can certify robustness against non-$\ell_p$ changes, such as visible color shifts, for which the $\ell_2$-norm of the perturbation in the image space will be very large. Another line of work proves generalization bounds with for divergence-based measures of distribution shift (Ben-David et al., 2006; Zhao et al., 2019; Mehra et al., 2021; Weber et al., 2022) like KL-divergence, total variation distance and Hellinger distance. Divergence measures between two distributions become arbitrarily large (e.g. KL-divergence becomes infinity) or attain their maximal value (e.g. total variation and Hellinger distances become equal to one) when their supports do not coincide. This drawback makes them unsuitable for measuring out-of-distribution data shifts which by definition have non-overlapping support. Wasserstein distance, on the other hand, captures the spatial separation of two distributions and produces a more meaningful measure of the distance even when their supports are disjoint.

## 3 PRELIMINARIES AND NOTATIONS

Let $\mathcal{D}$ be the data distribution representing a machine learning task over an input space $\mathcal{X}$ and an output space $\mathcal{Y}$. We define a distribution shift as a covariate shift that only changes the distribution of the input element in samples $(x, y) \in \mathcal{X} \times \mathcal{Y}$ drawn from $\mathcal{D}$ and leaves the output element unchanged, i.e., $(x, y)$ changes to $(\tilde{x}, y)$ under the shift. Given a distance function $d_{\mathcal{X}} : \mathcal{X} \times \mathcal{X} \to \mathbb{R}_{\geq 0}$ over the input space, we define the following distance function between two tuples $\tau_1 = (x_1, y_1)$ and $\tau_2 = (x_2, y_2)$ to capture the above shift:

$$d(\tau_1, \tau_2) = \begin{cases} d_{\mathcal{X}}(x_1, x_2) & \text{if } y_1 = y_2 \\ \infty & \text{otherwise.} \end{cases} \tag{1}$$

Let $\tilde{\mathcal{D}}$ denote a shift in the original data distribution $\mathcal{D}$ such that the Wasserstein distance under $d$ between $\mathcal{D}$ and $\tilde{\mathcal{D}}$ is bounded by $\epsilon$ (i.e., $W_1^d(\mathcal{D}, \tilde{\mathcal{D}}) \leq \epsilon$). Define the set of all joint probability distributions with marginals $\mu_{\mathcal{D}}$ and $\mu_{\tilde{\mathcal{D}}}$ as follows:

$$\Gamma(\mathcal{D}, \tilde{\mathcal{D}}) = \left\{ \gamma \text{ s.t. } \int_{\mathcal{X} \times \mathcal{Y}} \gamma(\tau_1, \tau_2) d\tau_2 = \mu_{\mathcal{D}}(\tau_1) \text{ and } \int_{\mathcal{X} \times \mathcal{Y}} \gamma(\tau_1, \tau_2) d\tau_1 = \mu_{\tilde{\mathcal{D}}}(\tau_2) \right\}.$$

The Wasserstein bound implies that there exists an element $\gamma^* \in \Gamma(\mathcal{D}, \tilde{\mathcal{D}})$ such that

$$\mathbb{E}_{(\tau_1, \tau_2) \sim \gamma^*}[d(\tau_1, \tau_2)] \leq \epsilon. \tag{2}$$

Let $\mathcal{S} : \mathcal{X} \to \Delta(\mathcal{X})$ be a function mapping each element $x \in \mathcal{X}$ to a smoothing distribution $\mathcal{S}(x)$, where $\Delta(\mathcal{X})$ is the set of all probability distributions over $\mathcal{X}$. For example, smoothing with an isometric Gaussian noise distribution with variance $\sigma^2$ can be denoted as $\mathcal{S}(x) = \mathcal{N}(x, \sigma^2 I)$. Let the total variation between the smoothing distributions at two points $x_1$ and $x_2$ be bounded by a concave increasing function $\psi$ of the distance between them, i.e.,

$$\mathsf{TV}(\mathcal{S}(x_1), \mathcal{S}(x_2)) \leq \psi(d_{\mathcal{X}}(x_1, x_2)). \tag{3}$$

For example, when the distance function $d$ is the $\ell_2$-norm of the difference of $x_1$ and $x_2$, and the smoothing distribution is an isometric Gaussian $\mathcal{N}(0, \sigma^2 I)$ with variance $\sigma^2$, $\psi(\cdot) = \mathrm{erf}(\cdot / 2\sqrt{2}\sigma)$ is a valid upper bound on the above total variation that is concave in the positive domain (see Appendix G for more examples).

Consider a function $h : \mathcal{X} \times \mathcal{Y} \to [0, 1]$ that represents the performance (e.g., accuracy) of a model $\mu$ over all possible input-output pairs. For example, in the case of a classifier $\mu : \mathcal{X} \to \mathcal{Y}$ that maps inputs from space $\mathcal{X}$ to a class label in $\mathcal{Y}$, $h(x, y) := \mathbf{1}\{\mu(x) = y\}$ could indicate whether the prediction of $\mu$ on $x$ matches the desired output label $y$ or not. Another example could be that of segmentation/detection tasks, where $y$ represents a region on an input image $x$. Then, $h(x, y) := \mathrm{IoU}(\mu(x), y)$[1] could represent the overlap between the predicted regions $\mu(x)$ and the ground truth $y$. The overall accuracy of the model $\mu$ under $\mathcal{D}$ is then given by $\mathbb{E}_{(x,y) \in \mathcal{D}}[h(x, y)]$. Now, define a robust model $\bar{\mu}(x) = \mu(x')$ where $x' \sim \mathcal{S}(x)$ which simply applies the base model $\mu$ on a randomized version of the input $x$ sampled from a smoothing distribution $\mathcal{S}(x)$. Our goal is to bound the difference in the expected performance of the robust model between the original distribution $\mathcal{D}$ and the shifted distribution $\tilde{\mathcal{D}}$. Let $\bar{h}$ be the performance function for the robust model $\bar{\mu}$ defined as

$$\bar{h}(x, y) = \mathbb{E}_{x' \sim \mathcal{S}(x)}[h(x', y)]. \tag{4}$$

Then, the accuracy of the robust model $\bar{\mu}$ under $\mathcal{D}$ is given by $\mathbb{E}_{(x,y) \in \mathcal{D}}[\bar{h}(x, y)]$. Our result in Theorem 1 bounds the difference between the expectation of $\bar{h}$ under $\mathcal{D}$ and $\tilde{\mathcal{D}}$ with $\psi(\epsilon)$.

## 3.1 Parameterized Transformations

We apply our distributional certificates to produce guarantees on the accuracy of an image classifier under natural transformations such as color shifts, hue shifts and changes in brightness and saturation. We model each transformation as a function $\mathcal{T} : \mathcal{X} \times P \to \mathcal{X}$ over the image space $\mathcal{X}$ and a parameter space $P$. It takes an image $x \in \mathcal{X}$ and a parameter vector $\theta \in P$ as inputs and outputs a transformed image $x' = \mathcal{T}(x, \theta) \in \mathcal{X}$. An example of such a transformation could be a color shift in an RGB image produced by scaling the intensities in the red, green and blue channels $x = (\{x_{ij}^R\}, \{x_{ij}^G\}, \{x_{ij}^B\})$ defined as $\mathsf{CS}(x, \theta) = (2^{\theta_R}\{x_{ij}^R\}, 2^{\theta_G}\{x_{ij}^G\}, 2^{\theta_B}\{x_{ij}^B\})/\mathsf{MAX}$ for a tuple $\theta = (\theta_R, \theta_G, \theta_B)$, where $\mathsf{MAX}$ is the maximum of all the RGB values after scaling. Additive perturbations in the input space can also be captured as parameterized transformations, e.g., $\mathsf{VT}(x, \theta) = x + \theta$. We assume that the transformation returns $x$ if the parameters are all zero, i.e., $\mathcal{T}(x, 0) = x$ and that the composition of two transformations with parameters $\theta_1$ and $\theta_2$ is a transformation with parameters $\theta_1 + \theta_2$ (additive composability), i.e.,

$$\mathcal{T}(\mathcal{T}(x, \theta_1), \theta_2) = \mathcal{T}(x, \theta_1 + \theta_2). \tag{5}$$

---

[1]IoU stands for Intersection over Union.

Given a norm $\| \cdot \|$ in the parameter space $P$, we define a distance function in the input space $\mathcal{X}$ as follows:

$$d_{\mathcal{T}}(x_1, x_2) = \begin{cases} \min\{\|\theta\| \mid \mathcal{T}(x_1, \theta) = x_2\} & \text{if } \exists \theta \text{ s.t. } \mathcal{T}(x_1, \theta) = x_2 \\ \infty & \text{otherwise.} \end{cases} \quad (6)$$

Now, define a smoothing distribution $\mathcal{S}(x) = \mathcal{T}(x, \mathcal{Q}(0))$ for some distribution $\mathcal{Q}$ in the parameter space of $\mathcal{T}$ such that $\forall \theta \in P, \mathcal{Q}(\theta) = \theta + \mathcal{Q}(0)$ is the distribution of $\theta + \delta$ where $\delta \sim \mathcal{Q}(0)$, and $\mathsf{TV}(\mathcal{Q}(0), \mathcal{Q}(\theta)) \leq \psi(\|\theta\|)$ for a concave function $\psi$. For example, $\mathcal{Q}(\cdot) = \mathcal{N}(\cdot, \sigma^2 I)$ satisfies these properties for $\psi(\cdot) = \mathrm{erf}(\cdot/2\sqrt{2}\sigma)$. Then, the following lemma holds (proof in Appendix B):

**Lemma 1.** *For two points $x_1, x_2 \in \mathcal{X}$ such that $d_{\mathcal{T}}(x_1, x_2)$ is finite,*

$$\mathsf{TV}(\mathcal{S}(x_1), \mathcal{S}(x_2)) \leq \psi(d_{\mathcal{T}}(x_1, x_2)).$$

## 4 CERTIFIED DISTRIBUTIONAL ROBUSTNESS

In this section, we state our main theoretical result which shows that the difference in the expectation of the performance function $\bar{h}$ of the robust model (equation (4)) under the original distribution $\mathcal{D}$ and any shifted distribution $\tilde{\mathcal{D}}$ within a Wasserstein distance of $\epsilon$ from $\mathcal{D}$ is bounded by $\psi(\epsilon)$, where $\psi$ is the concave upper bound on the total variation between the smoothing distributions at two points $x_1$ and $x_2$ as defined in condition (3).

**Theorem 1.** *Given a function $h : \mathcal{X} \times \mathcal{Y} \to [0, 1]$, define its smoothed version as $\bar{h}(x, y) = \mathbb{E}_{x' \sim \mathcal{S}(x)}[h(x', y)]$. Then,*

$$\forall \, \tilde{\mathcal{D}} \text{ s.t. } W_1^d(\mathcal{D}, \tilde{\mathcal{D}}) \leq \epsilon, \quad \left| \mathbb{E}_{(x_1, y_1) \sim \mathcal{D}}[\bar{h}(x_1, y_1)] - \mathbb{E}_{(x_2, y_2) \sim \tilde{\mathcal{D}}}[\bar{h}(x_2, y_2)] \right| \leq \psi(\epsilon).$$

We defer the proof to Appendix A. Note that this certificate does not require us to compute the Wasserstein distance between $\mathcal{D}$ and $\tilde{\mathcal{D}}$. Given a value for $\epsilon$, it holds for *all* distributions $\tilde{\mathcal{D}}$ such that $W_1^d(\mathcal{D}, \tilde{\mathcal{D}}) \leq \epsilon$. Our certified guarantees hold for the entire input distribution (potentially continuous) and not just for a finite set of samples. The intuition behind the above bound is that if the overlap between the smoothing distributions between two individual points does not decrease rapidly with the distance between them, then the overlap between $\mathcal{D}$ and $\tilde{\mathcal{D}}$ augmented with the smoothing distribution is high when the Wasserstein distance between them is small. The key observation here is that the total variation of the individual smoothing distributions can be upper bounded by a convex function $\psi$ and this upper bound can then be generalised over the entire distribution using Jensen's inequality. The above guarantee implies that for any distribution $\tilde{\mathcal{D}}$ that is within a Wasserstein distance of $\epsilon$ from the original distribution $\mathcal{D}$, the accuracy of the model under $\tilde{\mathcal{D}}$ can be bounded as $\mathbb{E}_{(x_2, y_2) \sim \tilde{\mathcal{D}}}[\bar{h}(x_2, y_2)] \geq \mathbb{E}_{(x_1, y_1) \sim \mathcal{D}}[\bar{h}(x_1, y_1)] - \psi(\epsilon)$.

### 4.1 COMPUTING THE CERTIFICATE AND EMPIRICAL EVALUATIONS

Given a target Wasserstein bound $\epsilon$ and an appropriate function $\psi$, we simply need to calculate the expected performance of the robust model over the original distribution $\mathcal{D}$, i.e., $\mathbb{E}_{(x_1, y_1) \sim \mathcal{D}}[\bar{h}(x_1, y_1)]$. Since we only have sample access to the original distribution $\mathcal{D}$, we estimate the expected performance on $\mathcal{D}$, i.e. $\mathbb{E}_{(x_1, y_1) \sim \mathcal{D}}[\bar{h}(x_1, y_1)]$, using a finite number of samples. In our experiments, we compute a high-confidence lower bound of this quantity using the Clopper-Pearson method (Clopper & Pearson, 1934) that holds with $1 - \alpha$ probability, for some $\alpha > 0$ (usually 0.001). Note that although we calculate the bound with a finite number of samples from the distribution $\mathcal{D}$, this lower bound holds for the expectation over the *entire* distribution and not just for the samples. See Appendix C for pseudocodes of the prediction and certification steps.

To compare our certified guarantees against the empirical performance of an undefended model under distribution shifts, we design shifted distributions using natural and adversarial transformations on the original distribution. We ensure that the constructed distribution shift is within the desired Wasserstein distance using two methods:

1. By construction: We analytically guarantee beforehand that the applied transformation does not exceed the Wasserstein bound. For example, in Figure 2, we report the empirical performance of the base models under distribution shifts constructed by adding a noise vector

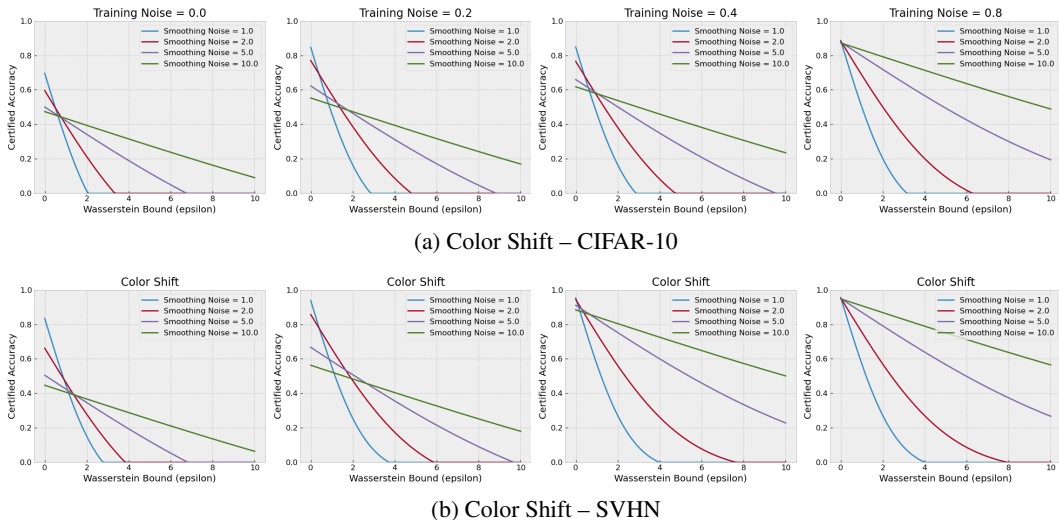

(a) Color Shift – CIFAR-10

(b) Color Shift – SVHN

Figure 3: Certified accuracy under color shifts for (a) CIFAR-10 and (b) SVHN. Each plot corresponds to a particular training noise and each curve corresponds to a particular smoothing noise.

      from a fixed distribution like a Gaussian distribution of a certain variance in the transformation space (see Appendix I).

2. By estimation: We compute a high-confidence bound on the average perturbation added to a finite number of samples to bound the Wasserstein distance. For example, in Section 6, when reporting the undefended baseline performance, we measure $\mathbb{E}[\|\mathrm{Adv}(x) - x\|_2]$ on the test set, and use Hoeffding's inequality to derive from this a 99% confidence upper bound on the true, population expectation $\mathbb{E}_{x \sim \mathcal{D}}[\|\mathrm{Adv}(x) - x\|_2]$. By Equation 8, this is a (high-probability) upper bound on the Wasserstein distance of the distribution shift.

In the following sections, we apply our main theoretical result to obtain certified robustness guarantees against several different distribution shifts – natural shifts, unlearnable distributions and adversarial shifts. We experiment on two image classification datasets, namely CIFAR-10 (Krizhevsky et al.) and SVHN (Netzer et al., 2011), and observe that the our certificates can obtain meaningful performance guarantees and exhibit similar trends for both datasets.

## 5    Certified Accuracy against Natural Transformations

We certify the accuracy of a ResNet-110 model and a ResNet-20 model trained on CIFAR-10 and SVHN images respectively under three types of transformations: color shifts, hue shifts and variation in brightness and saturation (SV shift). We train our models with varying levels of noise in the transformation space and evaluate their certified performance using smoothing distributions of different standard deviations. For color and SV shifts, we show how the certified accuracy varies as a function of the Wasserstein distance as we change the training and smoothing noise. For hue shift, we use a smoothing distribution (with fixed noise level) that is invariant to rotations in hue space because of which the certified accuracy remains constant with respect to the corresponding Wasserstein distance. We train the ResNet-110 models for 90 epochs which takes a few hours on a single NVIDIA GeForce RTX 2080 Ti GPU and the ResNet-20 models for 40 epochs which takes around twenty minute on the same GPU. Once the models have been trained, computing the distribution level Wasserstein certificates using $10^5$ samples with 99.9% confidence takes only about 25 seconds for each model.

### 5.1    Color Shifts

Denote an RGB image $x$ as an $H \times W$ array of pixels where the red, green and blue components of the pixel in the $i$th row and $j$th column are given by the tuple $x_{ij} = (r, g, b)_{ij}$. Let $r_{\max}, g_{\max}$ and $b_{\max}$ be the maximum values of the red, green and blue channels, respectively. Assume that the RGB values are in the interval $[0, 1]$ normalized such that the maximum over all intensity values is one, i.e., $\max(r_{\max}, g_{\max}, b_{\max}) = 1$. Define a color shift of the image $x$ for a parameter vector $\theta \in \mathbb{R}^3$ as

$$\mathsf{CS}(x, \theta) = \left\{ \frac{(2^{\theta_R} r, 2^{\theta_G} g, 2^{\theta_B} b)_{ij}}{\max(2^{\theta_R} r_{\max}, 2^{\theta_G} g_{\max}, 2^{\theta_B} b_{\max})} \right\}^{H \times W}$$

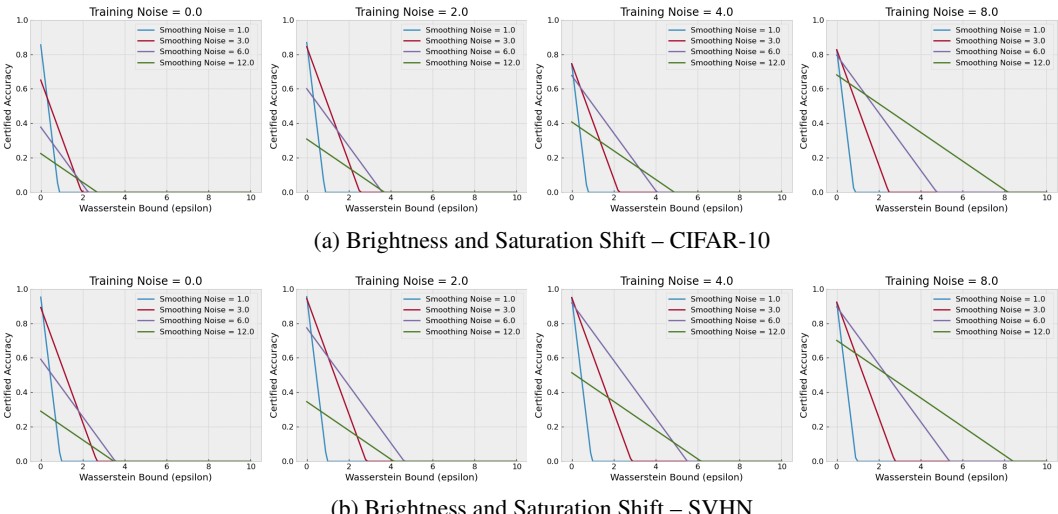

(a) Brightness and Saturation Shift – CIFAR-10

(b) Brightness and Saturation Shift – SVHN

Figure 4: Certified accuracy under brightness and saturation changes for (a) CIFAR-10 and (b) SVHN images. Each plot corresponds to a particular training noise and each curve corresponds to a particular smoothing noise.

which scales the intensities of each channel by the corresponding component of $\theta$ raised to the power of two and then normalizes the scaled image so that the maximum intensity is one. For example, $\theta = (1, -1, 0)$ would first double all the red intensities, halve the green intensities and leave the blue intensities unchanged, and then, normalize the image so that the maximum intensity value over all the channels is equal to one. The above transformation can be shown to satisfy the additive composability property in condition (5). See Appendix H for a proof.

Given an image $x$, we define a smoothing distribution around $x$ in the parameter space as $\mathsf{CS}(x, \delta)$ where $\delta \sim \mathcal{N}(0, \sigma^2 I_{3\times3})$. Define the distance function $d_{\mathsf{CS}}$ as described in (6) using the $\ell_2$-norm in the parameter space. For a distribution $\tilde{\mathcal{D}}$ within a Wasserstein distance of $\epsilon$ from the original distribution $\mathcal{D}$, the performance of the smoothed model on $\tilde{\mathcal{D}}$ can be bounded as $\mathbb{E}_{(x_2,y_2)\sim\tilde{\mathcal{D}}}[\bar{h}(x_2, y_2)] \geq \mathbb{E}_{(x_1,y_1)\sim\mathcal{D}}[\bar{h}(x_1, y_1)] - \mathrm{erf}(\epsilon/2\sqrt{2}\sigma)$. Figure 3 plots the certified accuracy under color shift with respect to the Wasserstein bound $\epsilon$ for different values of training and smoothing noise. In Appendix K, we consider a smoothing distribution that randomly picks one color channel achieving a constant certified accuracy of 87.1% with respect to $\epsilon$.

## 5.2 BRIGHTNESS AND SATURATION CHANGES

Define the following transformation in the HSV space of an image that shifts the mean of the saturation (S) and brightness (V) values for each pixel by a certain amount:

$$\mathsf{SV}(x, \theta) = \left\{ \left( h, \frac{s + (2^{\theta_S} - 1)s_{\mathrm{mean}}}{\mathsf{MAX}}, \frac{v + (2^{\theta_V} - 1)v_{\mathrm{mean}}}{\mathsf{MAX}} \right)_{ij} \right\}^{H \times W}$$

where $s_{\mathrm{mean}}, s_{\mathrm{max}}, v_{\mathrm{mean}}$ and $v_{\mathrm{max}}$ are the means and maximums of the saturation and brightness values respectively before the shift is applied and $\mathsf{MAX} = \max(s_{\mathrm{max}} + (2^{\theta_S} - 1)s_{\mathrm{mean}}, v_{\mathrm{max}} + (2^{\theta_V} - 1)v_{\mathrm{mean}})$ is the maximum of the brightness and saturation values after the shift. Similar to color shift , the $\mathsf{SV}$ transformation can also be shown to satisfy additive composability (Appendix H). Figure 4 plots the certified accuracy under saturation and brightness changes with respect to $\epsilon$ for different values of training and smoothing noise. The smoothing distribution is uniform in the range $[0, a]^2$ in the parameter space, the distance function is the $\ell_1$-norm and $\psi(\epsilon) = \min(\epsilon/a, 1)$.

## 6 POPULATION-LEVEL CERTIFICATES AGAINST ADVERSARIAL ATTACKS

In this section, we consider the $\ell_2$-distance in the image space to measure the Wasserstein distance instead of a parameterized transformation (see Appendix D for a detailed version). We use a pixel-space Gaussian smoothing distribution $\mathcal{S}(x) = \mathcal{N}(x, \sigma^2 I)$ to obtain robustness guarantees under this metric. To motivate this, consider an adversarial attacker $\mathrm{Adv} : \mathcal{X} \to \mathcal{X}$, which takes an image $x$ and computes perturbation $\mathrm{Adv}(x)$ to try and fool a model into misclassifying the input. If

$(x, y) \sim \mathcal{D}$, define $\tilde{\mathcal{D}}$ to be the distribution of the tuples $(\text{Adv}(x), y)$. Defining $d$ in 1 using $d_{\mathcal{X}} = \ell_2$, it is easy to show that:

$$W_1^d(\mathcal{D}, \tilde{\mathcal{D}}) \leq \mathbb{E}_{x \sim \mathcal{D}}[\|\text{Adv}(x) - x\|_2] \tag{7}$$

Results on CIFAR-10 are presented in Figure 5 and results on SVHN are available in Appendix D. For CIFAR-10, we use ResNet-110 models trained under noise from Cohen et al. (2019). The solid lines represent certified accuracies for different smoothing noises and the black dashed line represents the empirical performance of an undefended model under attack. For the undefended baseline, we give the performance of an undefended model against a *strategic* attacker, which first finds a minimal $\ell_2$ attack for each sample via (Carlini & Wagner, 2017). If this attack is too large in magnitude ($\ell_2 > a$ threshold $\gamma$), it instead chooses not to attack the sample. This "saves" the attack budget (i.e., the average attack magnitude and therefore the Wasserstein shift) for easier samples. The size of the Wasserstein shift can be adjusted by varying $\gamma$.

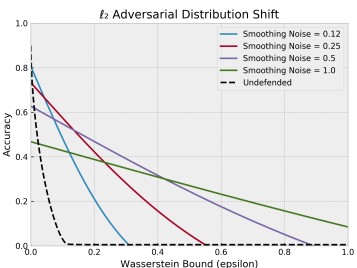

Figure 5: Distributional certificates against adversarial attacks on CIFAR-10.

## 7 HARDNESS RESULTS ON UNLEARNABILITY

In this section, we show that the pixel-space $\ell_2$-Wasserstein distributional robustness certificate shown above can also be applied to establish a hardness result in creating provably "unlearnable" datasets (Huang et al., 2021). These datasets contain "poisoned" samples which make any classifier trained on the released data achieve a high training and validation accuracy, but a low test accuracy on non-poisoned samples from the original data distribution. This technique has legitimate applications, such as protecting privacy by preventing one's personal data from being learned, but may also have malicious uses (e.g., a malicious actor could sell a useless classifier that nevertheless has good performance on a provided validation set.) We can view the "clean" data distribution as $\mathcal{D}$, and the distribution of the poisoned samples (i.e., the unlearnable distribution) as $\tilde{\mathcal{D}}$. If the magnitude of the perturbations is limited, Theorem 1 implies that the accuracy on $\mathcal{D}$ and $\tilde{\mathcal{D}}$ must be similar, implying that our algorithm is provably resistant to unlearnablility attacks, effectively establishing provable hardness results to create unlearnable datasets.

In order to apply our guarantees, we make a few modifications to the attack proposed in Huang et al. (2021). First, we bound each poisoning perturbation on the released dataset to within an $\epsilon$-radius $\ell_2$ ball, rather than an $\ell_\infty$ ball. From Equation 8, this ensures that $W_1^d(\mathcal{D}, \tilde{\mathcal{D}}) \leq \epsilon$. Second, we consider an "offline" version of the attack. In the original attack (Huang et al., 2021), perturbations for the entire dataset are optimized simultaneously with a proxy classifier model in an iterative manner. This makes the perturbations applied to each sample non-I.I.D., (they may depend on each other through proxy-model parameters) which makes deriving generalizable guarantees for it difficult.

However, this simultaneous proxy-model training and poisoning may not always represent a realistic threat model. In particular, an actor releasing "unlearnable" data at scale may not be able to constantly update the proxy model being used. For example, consider an "unlearnability" module in a camera, which would make photos unusable as training data. Because the camera itself has access to only a small number of photographs, such a module would likely rely on a fixed, pre-trained proxy classifier model to create the poisoning perturbations. To model this, we consider a threat model where the proxy classifier is first optimized using an unreleased dataset: the released "unlearnable" samples are then perturbed independently using this fixed proxy model. We see in Figure 6 that our modified attack is still highly effective at making data unlearnable, as shown by the high validation and low test accuracy of the undefended baseline.

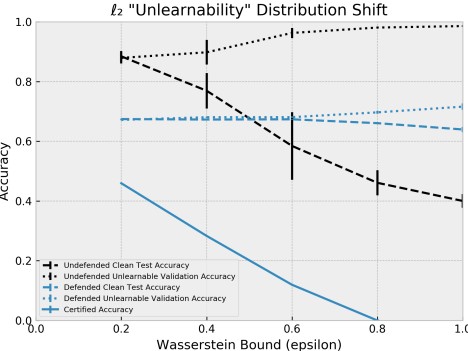

Figure 6: Distributional certificates for unlearnable datasets on CIFAR-10. The smoothing noise used is $0.4$. Results for other values are reported in the appendix.

ACKNOWLEDGEMENTS

This project was supported in part by Meta grant 23010098, NSF CAREER AWARD 1942230, HR001119S0026 (GARD), ONR YIP award N00014-22-1-2271, Army Grant No. W911NF2120076, a capital one grant, NIST 60NANB20D134, and the NSF award CCF2212458.

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

## A    PROOF OF THEOREM 1

**Statement.** *Given a function $h : \mathcal{X} \times \mathcal{Y} \to [0,1]$, define its smoothed version as $\bar{h}(x,y) = \mathbb{E}_{x' \sim \mathcal{S}(x)}[h(x',y)]$. Then,*

$$\left| \mathbb{E}_{(x_1,y_1) \sim \mathcal{D}}[\bar{h}(x_1,y_1)] - \mathbb{E}_{(x_2,y_2) \sim \tilde{\mathcal{D}}}[\bar{h}(x_2,y_2)] \right| \leq \psi(\epsilon).$$

*Proof.* Let $\tau_1 = (x_1, y_1)$ and $\tau_2 = (x_2, y_2)$ denote the input-output tuples sampled from $\mathcal{D}$ and $\tilde{\mathcal{D}}$ respectively. Then, for the joint distribution $\gamma^* \in \Gamma(\mathcal{D}, \tilde{\mathcal{D}})$ in (2), we have

$$\mathbb{E}_{\tau_1 \sim \mathcal{D}}[\bar{h}(\tau_1)] = \mathbb{E}_{(\tau_1,\tau_2) \sim \gamma^*}[\bar{h}(\tau_1)] \quad \text{and} \quad \mathbb{E}_{\tau_2 \sim \tilde{\mathcal{D}}}[\bar{h}(\tau_2)] = \mathbb{E}_{(\tau_1,\tau_2) \sim \gamma^*}[\bar{h}(\tau_2)].$$

This is because when $(\tau_1, \tau_2)$ is sampled from the joint distribution $\gamma^*$, $\tau_1$ and $\tau_2$ individually have distributions $\mathcal{D}$ and $\tilde{\mathcal{D}}$ respectively. Also, since the expected distance between $\tau_1 = (x_1, y_1)$ and $\tau_2 = (x_2, y_2)$ is finite, the output elements of the sampled tuples must be the same, i.e. $y_1 = y_2 = y$ (say). See lemma 2 below. Then,

$$\begin{aligned} \left| \mathbb{E}_{(x_1,y_1) \sim \mathcal{D}}[\bar{h}(x_1,y_1)] - \mathbb{E}_{(x_2,y_2) \sim \tilde{\mathcal{D}}}[\bar{h}(x_2,y_2)] \right| \\ = \left| \mathbb{E}_{\tau_1 \sim \mathcal{D}}[\bar{h}(\tau_1)] - \mathbb{E}_{\tau_2 \sim \tilde{\mathcal{D}}}[\bar{h}(\tau_2)] \right| \\ = \left| \mathbb{E}_{(\tau_1,\tau_2) \sim \gamma^*}[\bar{h}(\tau_1) - \bar{h}(\tau_2)] \right| \\ \leq \mathbb{E}_{(\tau_1,\tau_2) \sim \gamma^*}[|\bar{h}(\tau_1) - \bar{h}(\tau_2)|]. \end{aligned}$$

Now, from definition (4) and for $i = 1$ and $2$,

$$\bar{h}(\tau_i) = \bar{h}(x_i, y) = \mathbb{E}_{x_i' \sim \mathcal{S}(x_i)}[h(x_i', y)] = \mathbb{E}_{x_i' \sim \mathcal{S}(x_i)}[g(x_i')]$$

can be expressed as the expected value of a function $g : \mathcal{X} \to [0,1]$ under distribution $\mathcal{S}(x_i)$. Without loss of generality, assume $\mathbb{E}_{x_1' \sim \mathcal{S}(x_1)}[g(x_1')] \geq \mathbb{E}_{x_2' \sim \mathcal{S}(x_2)}[g(x_2')]$. Then,

$$\left| \mathbb{E}_{x_1' \sim \mathcal{S}(x_1)}[g(x_1')] - \mathbb{E}_{x_2' \sim \mathcal{S}(x_2)}[g(x_2')] \right|$$

$$= \int_{\mathcal{X}} g(x)\mu_1(x)dx - \int_{\mathcal{X}} g(x)\mu_2(x)dx$$

$$(\mu_1 \text{ and } \mu_2 \text{ are the PDFs of } \mathcal{S}(x_1) \text{ and } \mathcal{S}(x_1))$$

$$= \int_{\mathcal{X}} g(x)(\mu_1(x) - \mu_2(x))dx$$

$$= \int_{\mu_1 > \mu_2} g(x)(\mu_1(x) - \mu_2(x))dx - \int_{\mu_2 > \mu_1} g(x)(\mu_2(x) - \mu_1(x))dx$$

$$\leq \int_{\mu_1 > \mu_2} \max_{x' \in \mathcal{X}} g(x')(\mu_1(x) - \mu_2(x))dx - \int_{\mu_2 > \mu_1} \min_{x' \in \mathcal{X}} g(x')(\mu_2(x) - \mu_1(x))dx$$

$$\leq \int_{\mu_1 > \mu_2} (\mu_1(x) - \mu_2(x))dz$$

$$(\text{since } \max_{x' \in \mathcal{X}} g(x') \leq 1 \text{ and } \min_{x' \in \mathcal{X}} g(x') \geq 0)$$

$$= \frac{1}{2} \int_{\mathcal{X}} |\mu_1(x) - \mu_2(x)|dx = \mathsf{TV}(\mathcal{S}(x_1), \mathcal{S}(x_2)).$$

$$(\text{since } \int_{\mu_1 > \mu_2} (\mu_1(x) - \mu_2(x))dx = \int_{\mu_2 > \mu_1} (\mu_2(x) - \mu_1(x))dx = \tfrac{1}{2} \int_{\mathcal{X}} |\mu_1(x) - \mu_2(x)|dx)$$

Thus, from (1) and (3), we have $|\bar{h}(\tau_1) - \bar{h}(\tau_2)| \leq \psi(d_{\mathcal{X}}(x_1, x_2)) = \psi(d(\tau_1, \tau_2))$, and therefore,

$$\left| \mathbb{E}_{(x_1, y_1) \sim \mathcal{D}}[\bar{h}(x_1, y_1)] - \mathbb{E}_{(x_2, y_2) \sim \tilde{\mathcal{D}}}[\bar{h}(x_2, y_2)] \right|$$

$$\leq \mathbb{E}_{(\tau_1, \tau_2) \sim \gamma^*}[\psi(d(\tau_1, \tau_2))]$$

$$\leq \psi\left( \mathbb{E}_{(\tau_1, \tau_2) \sim \gamma^*}[d(\tau_1, \tau_2)] \right). \qquad (\psi \text{ is concave, Jensen's inequality})$$

Hence, from (2) and since $\psi$ is non-decreasing, we have

$$\left| \mathbb{E}_{(x_1, y_1) \sim \mathcal{D}}[\bar{h}(x_1, y_1)] - \mathbb{E}_{(x_2, y_2) \sim \tilde{\mathcal{D}}}[\bar{h}(x_2, y_2)] \right| \leq \psi(\epsilon).$$

$\square$

**Lemma 2.** *Let* $\Omega = \{(\tau_1, \tau_2) \text{ s.t. } y_1 \neq y_2 \text{ where } \tau_1 = (x_1, y_1) \text{ and } \tau_2 = (x_2, y_2)\}$. *Then*

$$\mathbb{P}_{(\tau_1, \tau_2) \sim \gamma^*}[(\tau_1, \tau_2) \in \Omega] = 0.$$

*Proof.* Assume, for the sake of contradiction, that

$$\mathbb{P}_{(\tau_1, \tau_2) \sim \gamma^*}[(\tau_1, \tau_2) \in \Omega] \geq p$$

for some $p > 0$. From condition (2), we have

$$\mathbb{E}_{(\tau_1, \tau_2) \sim \gamma^*}[d(\tau_1, \tau_2)] \leq \epsilon.$$

By the law of total expectation

$$\mathbb{E}_{\gamma^*}[d(\tau_1, \tau_2)] = \mathbb{E}_{\gamma^*}[d(\tau_1, \tau_2) \mid (\tau_1, \tau_2) \in \Omega] \, \mathbb{P}_{\gamma^*}[(\tau_1, \tau_2) \in \Omega]$$
$$+ \mathbb{E}_{\gamma^*}[d(\tau_1, \tau_2) \mid (\tau_1, \tau_2) \notin \Omega] \, \mathbb{P}_{\gamma^*}[(\tau_1, \tau_2) \notin \Omega].$$

We replace $(\tau_1, \tau_2) \sim \gamma^*$ with just $\gamma^*$ in the subscripts for brevity. Since both summands are non-negative,

$$\mathbb{E}_{\gamma^*}[d(\tau_1, \tau_2) \mid (\tau_1, \tau_2) \in \Omega] \, \mathbb{P}_{\gamma^*}[(\tau_1, \tau_2) \in \Omega] \leq \epsilon.$$

Consider a real number $l > \epsilon/p$. Then, for any $(\tau_1, \tau_2) \in \Omega$, from definition (1) and because $y_1 \neq y_2$, $d(\tau_1, \tau_2) \geq l$. Therefore, $\mathbb{E}_{\gamma^*}[d(\tau_1, \tau_2) \mid (\tau_1, \tau_2) \in \Omega] \geq l$ and

$$l \, \mathbb{P}_{\gamma^*}[(\tau_1, \tau_2) \in \Omega] \leq \mathbb{E}_{\gamma^*}[d(\tau_1, \tau_2) \mid (\tau_1, \tau_2) \in \Omega] \, \mathbb{P}_{\gamma^*}[(\tau_1, \tau_2) \in \Omega]$$
$$l \, \mathbb{P}_{\gamma^*}[(\tau_1, \tau_2) \in \Omega] \leq \epsilon$$
$$\mathbb{P}_{\gamma^*}[(\tau_1, \tau_2) \in \Omega] \leq \epsilon/l < p,$$

which contradicts our initial assumption. $\square$

| **Algorithm 1** Prediction | **Algorithm 2** Certification |
|---|---|
| **Input:** Model $\mu$, input instance $x$. | **Input:** Accuracy function $h$, data distribution $\mathcal{D}$, Wasserstein bound $\epsilon$, integer $n$ and $\alpha > 0$. |
| **Output:** Robust prediction $y$. | **Output:** Certified accuracy for bound $\epsilon$. |
| Randomize input: $x' \sim \mathcal{S}(x)$. | sum $= 0$. |
| Evaluate model: $y = \mu(x')$. | **for** $i$ in $1 \dots n$ **do** |
| Return y. | $\quad$ Sample $(x, y) \sim \mathcal{D}$. |
| | $\quad$ Sample $x' \sim \mathcal{S}(x)$. |
| | $\quad$ Compute $h(x', y)$. |
| | $\quad$ sum $=$ sum $+ h(x', y)$ |
| | **end for** |
| | Compute $1 - \alpha$ confidence lower-bound $\underline{h}$ of $\mathbb{E}_{(x,y) \sim \mathcal{D}}[\bar{h}(x, y)]$ using sum and $n$. |
| | Return $\underline{h} - \psi(\epsilon)$. |

## B    PROOF OF LEMMA 1

**Statement.** *For two points $x_1, x_2 \in \mathcal{X}$ such that $d_{\mathcal{T}}(x_1, x_2)$ is finite,*

$$\mathsf{TV}(\mathcal{S}(x_1), \mathcal{S}(x_2)) \leq \psi(d_{\mathcal{T}}(x_1, x_2)).$$

*Proof.* Consider the $\theta$ for which $d_{\mathcal{T}}(x_1, x_2) = \|\theta\|$. Then, $\mathcal{T}(x_1, \theta) = x_2$.

$$
\begin{aligned}
\mathsf{TV}(\mathcal{S}(x), \mathcal{S}(x_2)) &= \mathsf{TV}(\mathcal{T}(x, \mathcal{Q}(0)), \mathcal{T}(z, \mathcal{Q}(0))) \\
&= \mathsf{TV}(\mathcal{T}(x, \mathcal{Q}(0)), \mathcal{T}(\mathcal{T}(x, \theta), \mathcal{Q}(0))) \\
&= \mathsf{TV}(\mathcal{T}(x, \mathcal{Q}(0)), \mathcal{T}(x, \theta + \mathcal{Q}(0))) \quad \text{(additive composability, equation (5))} \\
&= \mathsf{TV}(\mathcal{T}(x, \mathcal{Q}(0)), \mathcal{T}(x, \mathcal{Q}(\theta))). \quad \text{(definition of } \mathcal{Q})
\end{aligned}
$$

Let $A$ be the event in the space $M$ that maximizes the difference in the probabilities assigned to $A$ by $\mathcal{T}(x, \mathcal{Q}(0))$ and $\mathcal{T}(x, \mathcal{Q}(\theta))$. Let $u : P \to [0, 1]$ be a function that returns the probability (over the randomness of $\mathcal{T}$) of any parameter $\eta \in P$ being mapped to a point in $A$, i.e., $u(\eta) = \mathbb{P}\{\mathcal{T}(x, \eta) \in A\}$. For a deterministic transformation $\mathcal{T}$, $u$ is a 0/1 function. Then, the probabilities assigned by $\mathcal{T}(x, \mathcal{Q}(0))$ and $\mathcal{T}(x, \mathcal{Q}(\theta))$ to $A$ is equal to $\mathbb{E}_{\eta \sim \mathcal{Q}(0)}[u(\eta)]$ and $\mathbb{E}_{\eta \sim \mathcal{Q}(\theta)}[u(\eta)]$. Therefore,

$$
\begin{aligned}
\mathsf{TV}(\mathcal{S}(x), \mathcal{S}(x_2)) &= |\mathbb{E}_{\eta \sim \mathcal{Q}(0)}[u(\eta)] - \mathbb{E}_{\eta \sim \mathcal{Q}(\theta)}[u(\eta)]| \\
&\leq \mathsf{TV}(\mathcal{Q}(0), \mathcal{Q}(\theta)) \\
&\leq \psi(\|\theta\|) = \psi(d_{\mathcal{T}}(x_1, x_2)). \quad \text{(definition of } \mathcal{Q} \text{ and } d_{\mathcal{T}})
\end{aligned}
$$

$\square$

## C    PSEUDOCODE FOR PREDICTION AND CERTIFICATION

Algorithm 1 and Algorithm 2 describe the prediction and certification steps of our method.

## D    POPULATION-LEVEL CERTIFICATES AGAINST ADVERSARIAL ATTACKS

In this section, we consider the $\ell_2$-distance in the image space to measure the Wasserstein distance instead of a parameterized transformation. We use a pixel-space Gaussian smoothing distribution $\mathcal{S}(x) = \mathcal{N}(x, \sigma^2 I)$ to obtain robustness guarantees under this metric. To motivate this, consider an adversarial attacker $\text{Adv} : \mathcal{X} \to \mathcal{X}$, which takes an image $x$ and computes perturbation $\text{Adv}(x)$ to try and fool a model into misclassifying the input. If $(x, y) \sim \mathcal{D}$, define $\tilde{\mathcal{D}}$ to be the distribution of the tuples $(\text{Adv}(x), y)$. Defining $d$ in 1 using $d_{\mathcal{X}} = \ell_2$, it is easy to show that:

$$W_1^d(\mathcal{D}, \tilde{\mathcal{D}}) \leq \mathbb{E}_{x \sim \mathcal{D}}[\|\text{Adv}(x) - x\|_2] \tag{8}$$

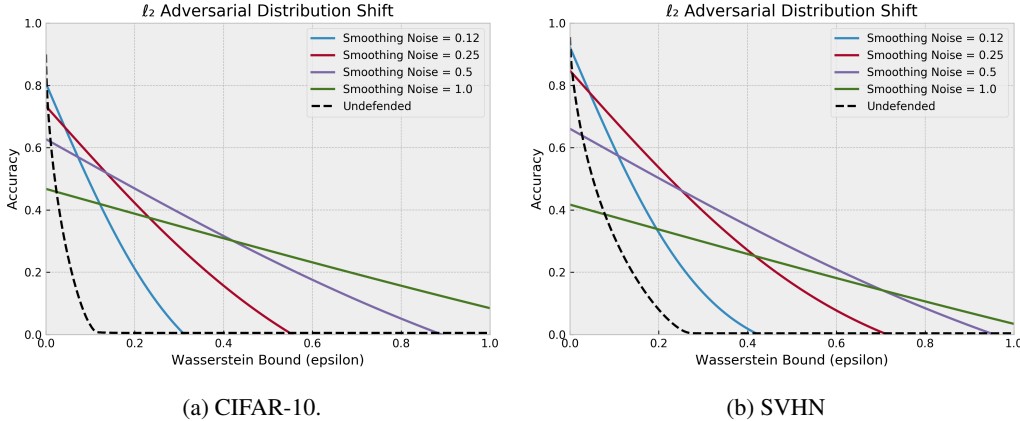

(a) CIFAR-10.                                        (b) SVHN

Figure 7: Distributional certificates against adversarial attacks on (a) CIFAR-10 and (b) SVHN. The solid lines represent certified accuracy of the robust models and the dashed lines represent the adversarial accuracy of undefended models.

So, if the *average* magnitude of perturbations induced by Adv is less than $\epsilon$ (i.e., $[\|\mathrm{Adv}(x) - x\|_2] < \epsilon$), then $W_1^d(\mathcal{D}, \tilde{\mathcal{D}}) < \epsilon$ which means that we can apply Theorem 1: the gap in the expected accuracy between $x \sim \mathcal{D}$ and $\mathrm{Adv}(x) \sim \tilde{\mathcal{D}}$ will be at most $\psi(\epsilon)$. Note that, under this threat model, Adv can be strategic in its use of the average perturbation "budget": if a certain point $x$ would require a very large perturbation to be misclassified, or is already misclassified, then $\mathrm{Adv}(x)$ can save the budget by simply returning $x$ and use it to attack a greater number of more vulnerable samples.

Note that our method differs from *sample-wise* certificates against $\ell_2$ adversarial attacks which use randomized smoothing, such as Cohen et al. (2019). Specifically, we use only one smoothing perturbation (and therefore only one forward pass) per sample. Our guarantees are on the overall accuracy of the classifier, not on the stability of any particular prediction. Finally, as discussed, our threat model is different, because we allow the adversary to strategically choose which samples to attack, with the certificate dependent on the *Wasserstein* magnitude of the *distributional* attack.

Results on CIFAR-10 and SVHN are presented in Figure 7a. For CIFAR-10, we use ResNet-110 models trained under noise from Cohen et al. (2019). For SVHN, we train our own models using the same training schedule as used for CIFAR-10 by Cohen et al. (2019), but we use ResNet-20 in place of ResNet-110. The solid lines represent certified accuracies for different smoothing noises and the black dashed line represents the empirical performance of an undefended model under attack. For the undefended baseline (on an undefended classifier $g$), we first apply a Carlini and Wagner $\ell_2$ attack to each sample $x$ (Carlini & Wagner, 2017), generating adversarial examples $x'$. Define this attack as the function $CW(\cdot)$, such that $x' = CW(x, y; g)$, where $y$ is the ground-truth label. (If the attack fails, $CW(x, y; g) = x$). We then define a *strategic* adversary $\mathrm{Adv}_\gamma$ that returns $CW(x, y; g)$ if $\|CW(x, y; g) - x\|_2 < \gamma$, otherwise it returns $x$.

By not attacking samples which would require the largest $\ell_2$ perturbations to cause misclassification, this attack efficiently balances maximizing misclassification rate with minimizing the Wasserstein distance between $\mathcal{D}$ and $\tilde{\mathcal{D}}$. The threshold parameter $\gamma$ controls the tradeoff between misclassifcation rate and the Wasserstein perturbation magnitude. Note that our attacker here is strategic in a way that takes more advantage of the distributional threat model than simply finding the minimal perturbation for each sample: by choosing to *not attack at all* on robust samples, it can successfully attack a larger number of more vulnerable samples. The 'Undefended' baseline in Figure 7a plots the accuracy on attacked test samples under adversary $\mathrm{Adv}_\gamma$, for a sweep of values of $\gamma$, against an upper bound on the Wasserstein distance, given by $\mathbb{E}_{x \sim \mathcal{D}}[\|\mathrm{Adv}_\gamma(x) - x\|_2]$. (In order to estimate $\mathbb{E}_{x \sim \mathcal{D}}[\|\mathrm{Adv}_\gamma(x) - x\|_2]$, we compute the average perturbation size over the test set and use Hoeffding inequality to upper-bound the population expectation with $99\%$ confidence.) We can observe a large gap between this undefended model performance under attack, and the certified robustness of our model, showing that our certificate is highly nonvacuous. In Appendix E, we include results to show the empirical robustness of the smoothed classifiers under an "adaptive" attack, based on the attack on sample-

# E  EMPIRICAL ATTACKS ON $\ell_2$-DISTRIBUTIONAL ROBUSTNESS.

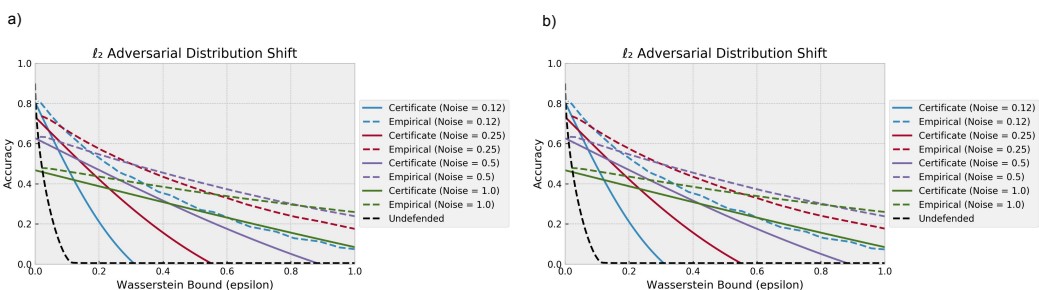

Figure 8: Adversarial attack on distributionally-smoothed classifiers, for CIFAR-10. For smoothed classifiers, we use the PGD attack described in is section; see Section 6 for details on the baseline. The dashed lines represent the empirical performance of the smoothed model for different noise levels. In plot (a), we use the loss function in Equation 9, while in (b) we use Equation 10.

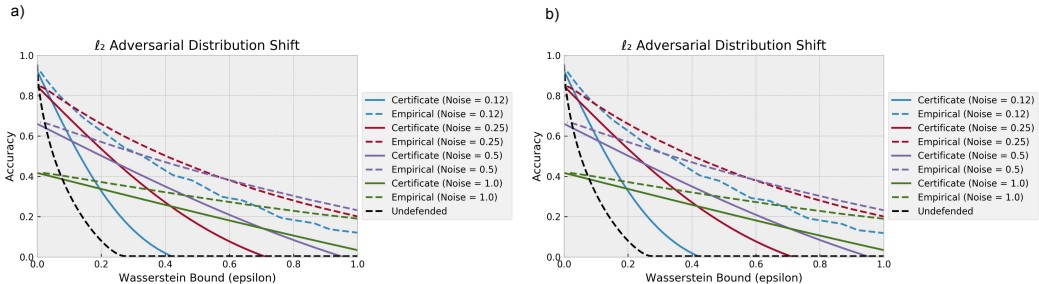

Figure 9: Adversarial attack on distributionally-smoothed classifiers, for SVHN. For smoothed classifiers, we use the PGD attack described in is section; see Section 6 for details on the baseline. The dashed lines represent the empirical performance of the smoothed model for different noise levels. In plot (a), we use the loss function in Equation 9, while in (b) we use Equation 10.

In this section, we describe an empirical attack on $\ell_2$-distributional smoothing. Our attack is based on the attack from Salman et al. (2019), and we use the code for PGD attack against smoothed classifiers from that work as a base, but there are a few considerations we must make.

First, while the goal of the attacker in Salman et al. (2019) is to change the output of a classifier that uses the *expected* logits, the goal in our case is to instead reduce the average classification accuracy of *each noise instance*. Concretely, Salman et al. (2019) uses an attacker loss function for each sample $x, y$ of the following form:

$$\max_\epsilon \mathcal{L}_{\text{Cross Ent.}} \left( \mathbb{E}_{\delta \sim \mathcal{N}(0, \sigma^2 I)} [\tilde{f}_\theta(x + \epsilon + \delta)], y \right) \tag{9}$$

Where we use $\tilde{f}$ to represent the SoftMax-ed logit function. However, because in our case, the classifier under attack is *not* $\mathbb{E}_{\delta \sim \mathcal{N}(0, \sigma^2 I)}[\tilde{f}_\theta(x + \epsilon + \delta)]$, but rather $\tilde{f}_\theta(x + \epsilon + \delta)$ itself, we instead considered the loss function:

$$\max_\epsilon \mathbb{E}_{\delta \sim \mathcal{N}(0, \sigma^2 I)} \left[ \mathcal{L}_{\text{Cross Ent.}} \left( \tilde{f}_\theta(x + \epsilon + \delta), y \right) \right] \tag{10}$$

Empirically, we find the choice of loss function to make very little difference: see Figures 8 and 9.

We also must consider how to correctly make the attacker "strategic": that is, how to allocate attack magnitude so as to attack most effectively while minimizing Wasserstein distance. This is more difficult than in the undefended case, because it is no longer true that for each sample $x$, we can identify the magnitude $\|CW(x, y; g) - x\|_2$ such that an attack of this magnitude is guaranteed to be successful, while a smaller attack is unsuccessful and hence is not attempted. Rather, for a given attack magnitude, there is instead a *probability of success*, over the distribution of $\delta$.

In order to deal with this, we perform PGD at a range of attack magnitudes, specifically $E = \{i/8 | i \in \{1, ..., 16\}\}$. Let $PGD_e(x, y; g)$ be the result of the attack at magnitude $e \in E$. We then define the adaptive attacker as:

$$\text{Adv}_\gamma(x) := PGD_{e*}(x, y; g) \tag{11}$$

Where:

$$e* := \max e \in E \text{ such that}$$

$$\frac{\mathbb{E}_\delta \left[ \mathcal{L}_{0/1} \left( \tilde{f}_\theta(PGD_e(x, y; g) + \delta), y \right) \right] - \mathbb{E}_\delta \left[ \mathcal{L}_{0/1} \left( \tilde{f}_\theta(x + \delta), y \right) \right]}{e} > \gamma \tag{12}$$

In other words, we use the largest attack such that the *increase in misclassification rate per unit attack magnitude* is above the threshold $\gamma$. If this is not the case for any $e \in E$, we elect not to attack, and set $\text{Adv}_\gamma(x) := x$. As was described in the main text for the baseline case, we sweep over a range of threshold values $\gamma$ when reporting results. When evaluating the expectations in Equation 12, we use a sample of 100 noise instances. However, once $e*$ is identified, we then use a *different* sample of 100 noise instances per training sample $x$ when reporting the final accuracy: this is to de-correlate the attack generation of $\text{Adv}_\gamma(x)$ with the evaluation of the attack. (However, noise instances are kept constant over the sweep of $\gamma$). When reporting results (the upper bounds of the Wasserstein distances), we use $e*$ as an upper bound on $\|PGD_{e*}(x, y; g) - x\|_2$, rather than using $\|PGD_{e*}(x, y; g) - x\|_2$ directly. Also, we upper bound the population expectation of $e*$ (and therefore of $\|PGD_{e*}(x, y; g) - x\|_2$) for each $\gamma$ with 99% confidence using the empirical expectation on the test set using a Hoeffding bound, using the fact that $0 \le e* \le \min(2, 1/\gamma)$.

Attack hyperparameters are taken from Salman et al. (2019): We use 20 attack steps, a step size of $e/10$, and use 128 noise instances when computing gradients. We evaluate using 10% of each dataset.

## F  EXPERIMENT DETAILS FOR SECTION 6

As mentioned, for the certified models, we use the released pre-trained ResNet110 models from Cohen et al. (2019) for CIFAR-10 and train ResNet20 models in a similar manner for SVHN, using the same level of Gaussian Noise for training and testing. For empirical results, we use the implementation of the $\ell_2$ Carlini and Wagner (Carlini & Wagner, 2017) attack provided by the IBM ART package (Nicolae et al., 2018) with default parameters (except for batch size which we set at 256 to increase processing speed.)

We also tested an alternative attack, which is still strategic but does not require that we measure the Wasserstein distance empirically. In this attack, we define $\text{Adv}'_\gamma$, that if $\|CW(x, y; g) - x\|_2 \le \gamma$ always returns $CW(x, y; g)$, and if $\|CW(x, y; g) - x\|_2 > \gamma$, instead returns $x$ with probability $1 - \frac{\gamma}{\|CW(x, y; g) - x\|_2}$. Note that in this case, the perturbation $\|\text{Adv}'_\gamma(x, y; g) - x\|_2$ is guaranteed to be less than or equal to $\gamma$ in expectation for all $x$, so $\gamma$ can be used as an upper bound on the Wasserstein distance. Results are shown in Figure 10.

## G  FUNCTION $\psi$ FOR DIFFERENT DISTRIBUTIONS

For an isometric Gaussian distribution $\mathcal{N}(0, \sigma^2 I)$,

$$\text{TV}(\mathcal{N}(0, \sigma^2 I), \mathcal{N}(\theta, \sigma^2 I)) = \text{erf}(\|\theta\|_2 / 2\sqrt{2}\sigma).$$

*Proof.* Due to the isometric symmetry of the Gaussian distribution and the $\ell_2$-norm, we may assume, without loss of generality, that $\mathcal{N}(\theta, \sigma^2 I)$ is obtained by shifting $\mathcal{N}(0, \sigma^2 I)$ only along the first

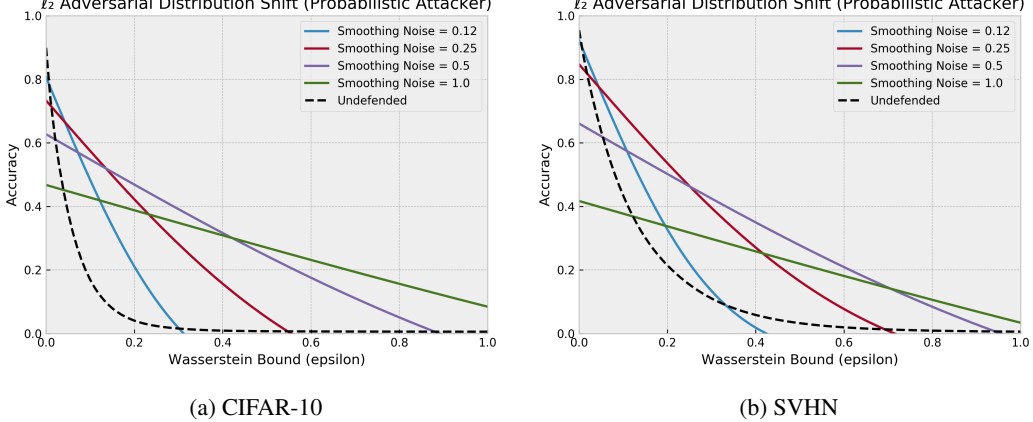

(a) CIFAR-10          (b) SVHN

Figure 10: Certified robustness to $\ell_2$ Wasserstein distributional attacks. The undefended baseline baseline is here attacked using the alternative attack formulation Adv′ described in Section F.

dimension. Therefore, the total variation of the two distributions is equal to the difference in the probability of a normal random variable with variance $\sigma^2$ being less than $\|\theta\|_2/2$ and $-\|\theta\|_2/2$, i.e., $\Phi(\|\theta\|_2/2\sigma) - \Phi(-\|\theta\|_2/2\sigma)$ where $\Phi$ is the standard normal CDF.

$$
\begin{aligned}
\mathsf{TV}(\mathcal{N}(0, \sigma^2 I), \mathcal{N}(\theta, \sigma^2 I)) &= \Phi(\|\theta\|_2/2\sigma) - \Phi(-\|\theta\|_2/2\sigma) \\
&= 2\Phi(\|\theta\|_2/2\sigma) - 1 \\
&= 2\left(\frac{1 + \mathrm{erf}(\|\theta\|_2/2\sqrt{2}\sigma)}{2}\right) - 1 \\
&= \mathrm{erf}(\|\theta\|_2/2\sqrt{2}\sigma).
\end{aligned}
$$

$\square$

For a uniform distribution $\mathcal{U}(\theta, b)$ between $\theta_i$ and $\theta_i + b$ in each dimension for $b \geq 0$ (as used for the SV shift transformations), $\mathsf{TV}(\mathcal{U}(0, b), \mathcal{U}(\theta, b)) \leq \|\theta\|_1/b$. When $\|\theta\|_1$ is constrained, the volume of the overlap between $\mathcal{U}(0, b)$ and $\mathcal{U}(\theta, b)$ is minimized when the shift is only along one dimension.

## H    ADDITIVE COMPOSABILITY OF NATURAL TRANSFORMATIONS

In this section, we prove that the natural transformation CS, HS and SV defined in the paper all satisfy the additive composability property in condition (5).

**Lemma 3.** *The transformation* CS *satisfies the additive composability property, i.e.,* $\forall x \in M, \theta_1, \theta_2 \in \mathbb{R}^3$,

$$
\mathsf{CS}(\mathsf{CS}(x, \theta_1), \theta_2) = \mathsf{CS}(x, \theta_1 + \theta_2).
$$

*Proof.* Let $x = \{(r, g, b)_{ij}\}^{H \times W}, x' = \{(r', g', b')_{ij}\}^{H \times W} = \mathsf{CS}(x, \theta_1)$ and $x'' = \{(r'', g'', b'')_{ij}\}^{H \times W} = \mathsf{CS}(x', \theta_2)$. We need to show that $x'' = \mathsf{CS}(x, \theta_1 + \theta_2)$. Let $r_{\max}, g_{\max}$ and $b_{\max}$ be the maximum values of the red, green and blue channels respectively of $x$ and $r'_{\max}, g'_{\max}$ and $b'_{\max}$ be the same for $x'$. From the definition of CS in Section 5.1, we have:

$$
r'_{ij} = \frac{2^{\theta_1^R} r_{ij}}{\mathsf{MAX}}, \quad g'_{ij} = \frac{2^{\theta_1^G} g_{ij}}{\mathsf{MAX}}, \quad b'_{ij} = \frac{2^{\theta_1^B} b_{ij}}{\mathsf{MAX}}
$$

$$
\text{and} \quad r''_{ij} = \frac{2^{\theta_2^R} r'_{ij}}{\mathsf{MAX}'}, \quad g''_{ij} = \frac{2^{\theta_2^G} g'_{ij}}{\mathsf{MAX}'}, \quad b''_{ij} = \frac{2^{\theta_2^B} b'_{ij}}{\mathsf{MAX}'}
$$

where $\mathsf{MAX} = \max(2^{\theta_1^R} r_{\max}, 2^{\theta_1^G} g_{\max}, 2^{\theta_1^B} b_{\max})$ and $\mathsf{MAX}' = \max(2^{\theta_2^R} r'_{\max}, 2^{\theta_2^G} g'_{\max}, 2^{\theta_2^B} b'_{\max})$. From the definition of $r'_{\max}$, we have:

$$
r'_{\max} = \max r'_{ij} = \max \frac{2^{\theta_1^R} r_{ij}}{\mathsf{MAX}} = \frac{2^{\theta_1^R} \max r_{ij}}{\mathsf{MAX}} = \frac{2^{\theta_1^R} r_{\max}}{\mathsf{MAX}}.
$$

Similarly,

$$g'_{\max} = \frac{2^{\theta_1^G} g_{\max}}{\mathsf{MAX}} \quad \text{and} \quad b'_{\max} = \frac{2^{\theta_1^B} b_{\max}}{\mathsf{MAX}}.$$

Therefore,

$$\mathsf{MAX}' = \frac{\max(2^{\theta_1^R + \theta_2^R} r_{\max}, 2^{\theta_1^G + \theta_2^G} g_{\max}, 2^{\theta_1^B + \theta_2^B} b_{\max})}{\mathsf{MAX}}.$$

Substituting $r'_{ij}$ and $\mathsf{MAX}'$ in the expression for $r''_{ij}$, we get:

$$r''_{ij} = \frac{2^{\theta_2^R} 2^{\theta_1^R} r_{ij}}{\mathsf{MAX}'\mathsf{MAX}} = \frac{2^{\theta_1^R + \theta_2^R} r_{ij}}{\max(2^{\theta_1^R + \theta_2^R} r_{\max}, 2^{\theta_1^G + \theta_2^G} g_{\max}, 2^{\theta_1^B + \theta_2^B} b_{\max})}.$$

Similarly,

$$g''_{ij} = \frac{2^{\theta_1^G + \theta_2^G} g_{ij}}{\max(2^{\theta_1^R + \theta_2^R} r_{\max}, 2^{\theta_1^G + \theta_2^G} g_{\max}, 2^{\theta_1^B + \theta_2^B} b_{\max})} \quad \text{and} \quad b''_{ij} = \frac{2^{\theta_1^B + \theta_2^B} b_{ij}}{\max(2^{\theta_1^R + \theta_2^R} r_{\max}, 2^{\theta_1^G + \theta_2^G} g_{\max}, 2^{\theta_1^B + \theta_2^B} b_{\max})}.$$

Hence, $x'' = \mathsf{CS}(x, \theta_1 + \theta_2)$. $\qquad\square$

**Lemma 4.** *The transformation* $\mathsf{SV}$ *satisfies the additive composability property, i.e.,* $\forall x \in M, \theta_1, \theta_2 \in \mathbb{R}_{\geq 0}^2$,

$$\mathsf{SV}(\mathsf{SV}(x, \theta_1), \theta_2) = \mathsf{SV}(x, \theta_1 + \theta_2).$$

*Proof.* Let $x = \{(h, s, v)_{ij}\}^{H \times W}$, $x' = \{(h, s', v')_{ij}\}^{H \times W} = \mathsf{SV}(x, \theta_1)$ and $x'' = \{(h, s'', v'')_{ij}\}^{H \times W} = \mathsf{SV}(x', \theta_2)$ in HSV format. We need to show that $x'' = \mathsf{SV}(x, \theta_1 + \theta_2)$. Let $s_{\mathrm{mean}}, s_{\max}, v_{\mathrm{mean}}$ and $v_{\max}$ be the means and maximums of the saturation and brightness values of $x$ and $s'_{\mathrm{mean}}, s'_{\max}, v'_{\mathrm{mean}}$ and $v'_{\max}$ be the same for $x'$. From the definition of $\mathsf{SV}$ in Section 5.2, we have:

$$s'_{ij} = \frac{s_{ij} + (2^{\theta_1^S} - 1)s_{\mathrm{mean}}}{\mathsf{MAX}}, \quad v'_{ij} = \frac{v_{ij} + (2^{\theta_1^V} - 1)v_{\mathrm{mean}}}{\mathsf{MAX}}$$

$$\text{and} \quad s''_{ij} = \frac{s'_{ij} + (2^{\theta_2^S} - 1)s'_{\mathrm{mean}}}{\mathsf{MAX}'}, \quad v''_{ij} = \frac{v'_{ij} + (2^{\theta_2^V} - 1)v'_{\mathrm{mean}}}{\mathsf{MAX}'}$$

where $\mathsf{MAX} = \max(s_{\max} + (2^{\theta_1^S} - 1)s_{\mathrm{mean}}, v_{\max} + (2^{\theta_1^V} - 1)v_{\mathrm{mean}})$ and $\mathsf{MAX}' = \max(s'_{\max} + (2^{\theta_2^S} - 1)s'_{\mathrm{mean}}, v'_{\max} + (2^{\theta_2^V} - 1)v'_{\mathrm{mean}})$. From the definitions of $s'_{\mathrm{mean}}$ and $s'_{\max}$, we have:

$$s'_{\mathrm{mean}} = \mathrm{mean}\, s'_{ij} = \mathrm{mean}\, \frac{s_{ij} + (2^{\theta_1^S} - 1)s_{\mathrm{mean}}}{\mathsf{MAX}} = \frac{\mathrm{mean}\, s_{ij} + (2^{\theta_1^S} - 1)s_{\mathrm{mean}}}{\mathsf{MAX}} = \frac{2^{\theta_1^S} s_{\mathrm{mean}}}{\mathsf{MAX}}$$

$$s'_{\max} = \max s'_{ij} = \max \frac{s_{ij} + (2^{\theta_1^S} - 1)s_{\mathrm{mean}}}{\mathsf{MAX}} = \frac{\max s_{ij} + (2^{\theta_1^S} - 1)s_{\mathrm{mean}}}{\mathsf{MAX}} = \frac{s_{\max} + (2^{\theta_1^S} - 1)s_{\mathrm{mean}}}{\mathsf{MAX}}.$$

Similarly,

$$v'_{\mathrm{mean}} = \frac{2^{\theta_1^V} v_{\mathrm{mean}}}{\mathsf{MAX}} \quad \text{and} \quad v'_{\max} = \frac{v_{\max} + (2^{\theta_1^V} - 1)v_{\mathrm{mean}}}{\mathsf{MAX}}.$$

Therefore,

$$\mathsf{MAX}' = \max(s'_{\max} + (2^{\theta_2^S} - 1)s'_{\mathrm{mean}}, v'_{\max} + (2^{\theta_2^V} - 1)v'_{\mathrm{mean}})$$

$$= \max(\frac{s_{\max} + (2^{\theta_1^S} - 1)s_{\mathrm{mean}} + (2^{\theta_2^S} - 1)2^{\theta_1^S} s_{\mathrm{mean}}}{\mathsf{MAX}}, v'_{\max} + (2^{\theta_2^V} - 1)v'_{\mathrm{mean}})$$

$$= \max(\frac{s_{\max} + (2^{\theta_1^S + \theta_2^S} - 1)s_{\mathrm{mean}}}{\mathsf{MAX}}, v'_{\max} + (2^{\theta_2^V} - 1)v'_{\mathrm{mean}})$$

$$= \max(s_{\max} + (2^{\theta_1^S + \theta_2^S} - 1)s_{\mathrm{mean}}, v_{\max} + (2^{\theta_1^V} - 1)v_{\mathrm{mean}} + (2^{\theta_2^V} - 1)2^{\theta_1^V} v_{\mathrm{mean}})/\mathsf{MAX}$$

$$= \max(s_{\max} + (2^{\theta_1^S + \theta_2^S} - 1)s_{\mathrm{mean}}, v_{\max} + (2^{\theta_1^V + \theta_2^V} - 1)v_{\mathrm{mean}})/\mathsf{MAX}.$$

Substituting $s'_{ij}$, $s'_{\text{mean}}$ and $\mathsf{MAX}'$ in the expression for $s''_{ij}$, we get:

$$s''_{ij} = \frac{s_{ij} + (2^{\theta_1^S} - 1)s_{\text{mean}} + (2^{\theta_2^S} - 1)2^{\theta_1^S}s_{\text{mean}}}{\mathsf{MAX}'\mathsf{MAX}}$$

$$= \frac{s_{ij} + (2^{\theta_1^S + \theta_2^S} - 1)s_{\text{mean}}}{\max(s_{\text{max}} + (2^{\theta_1^S + \theta_2^S} - 1)s_{\text{mean}}, v_{\text{max}} + (2^{\theta_1^V + \theta_2^V} - 1)v_{\text{mean}})}.$$

Similarly,

$$v''_{ij} = \frac{v_{ij} + (2^{\theta_1^V + \theta_2^V} - 1)v_{\text{mean}}}{\max(s_{\text{max}} + (2^{\theta_1^S + \theta_2^S} - 1)s_{\text{mean}}, v_{\text{max}} + (2^{\theta_1^V + \theta_2^V} - 1)v_{\text{mean}})}.$$

Hence, $x'' = \mathsf{SV}(x, \theta_1 + \theta_2)$. $\qquad\square$

## I  DETAILS FOR PLOTS IN FIGURE 2

The distribution shifts used to evaluate the empirical performance of the base models in Figure 2 have been generated by first sampling an image $x$ from the original distribution $\mathcal{D}$ and then randomly transforming it images from the original distribution by adding a noise in the corresponding transformation space. The Wasserstein bound of these shifts can be calculated by computing the expected perturbation size of the smoothing distribution. For example, the expected $\ell_2$-norm of a 3-dimensional Gaussian vector is given by $2\sqrt{2}\sigma/\sqrt{\pi}$ and expected $\ell_1$-norm a 2-dimensional vector sampled uniformly from $[0, b]^2$ is $b$.

The training and smoothing noise levels used for color shift, hue shift and SV shift are $(0.8, 10.0)$, $(180°, 180°)$ and $(8.0, 12.0)$ respectively.

## J  HUE SHIFT

Any RGB image can be alternatively represented in the HSV image format by mapping the $(r, g, b)$ tuple for each pixel to a point $(h, s, v)$ in a cylindrical coordinate system where the values $h, s$ and $v$ represent the hue, saturation and brightness (value) of the pixel. The mapping from the RGB coordinate to the HSV coordinate takes the $[0, 1]^3$ color cube and transforms it into a cylinder of unit radius and height. The hue values are represented as angles in $[0, 2\pi)$ and the saturation and brightness values are in $[0, 1]$. Define a hue shift of an $H \times W$ sized image $x$ by an angle $\theta \in [-\pi, \pi]$ in the HSV space that rotates each hue value by an angle $\theta$ and wraps it around to the $[0, 2\pi)$ range. In appendix J, we show that the certified accuracy under hue shifts does not depend on the Wasserstein distance of the shifted distribution and report the certified accuracies obtained by various base models trained under different noise levels.

Define a hue shift of an $H \times W$ sized image $x$ by an angle $\theta \in [-\pi, \pi]$ in the HSV space as:

$$\mathsf{HS}(x, \theta) = \left\{ (w(h + \theta), s, v)_{ij} \right\}^{H \times W}$$
$$\text{where} \quad w(x) = x - 2\pi \left\lfloor \frac{x}{2\pi} \right\rfloor$$

which rotates each hue value by an angle $\theta$ and wraps it around to the $[0, 2\pi)$ range. It is easy to show that this transformation satisfies additive composability in condition (5). The Wasserstein distance is defined using the corresponding distance function $d_{\mathsf{HS}}$ by taking the absolute value of the hue shift $|\theta|$.

**Lemma 5.** *The transformation* $\mathsf{HS}$ *satisfies the additive composability property, i.e.,* $\forall x \in M, \theta_1, \theta_2 \in [-\pi, \pi]$,

$$\mathsf{HS}(\mathsf{HS}(x, \theta_1), \theta_2) = \mathsf{HS}(x, \theta_1 + \theta_2).$$

*Proof.* Let $h$ be the hue value of the $(i, j)$th pixel of the image $x$. Since the transformation only affects the hue values, we ignore the other coordinates. The hue value after the transformation

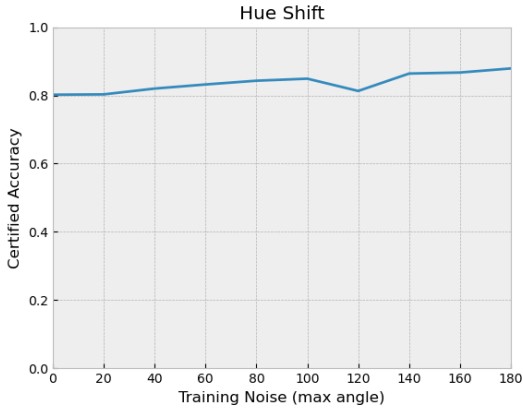

Figure 11: Certified accuracy under hue shift for different levels of training noise. Since, the certified accuracy remains constant with respect to the Wasserstein distance ($\epsilon$) of the shifted distribution, we plot the certified accuracy of models trained with different noise levels $\beta$.

$\mathsf{HS}(\mathsf{HS}(x, \theta_1), \theta_2)$ is given by

$$w(w(h + \theta_1) + \theta_2) = w\left(h + \theta_1 - 2\pi \left\lfloor \frac{h + \theta_1}{2\pi} \right\rfloor + \theta_2\right)$$

$\square$

Define a smoothing distribution that applies a random hue rotation $\delta$ sampled uniformly from the range $[-\pi, \pi]$. Since $\mathsf{HS}$ wraps the hue values around in the interval, the distributions of $h + \delta$ and $(h + \theta) + \delta$ for two hue values shifted by an angle $\theta$ are both uniform in $[0, 2\pi]$. Thus, the smoothing distribution for two hue shifted images is the same which implies that $\psi(d(x_1, x_2)) = 0$ whenever $d(x_1, x_2)$ is finite. Hence, from Theorem 1, we have $\mathbb{E}_{(x_2,y_2)\sim\tilde{\mathcal{D}}}[\bar{h}(x_2, y_2)] \geq \mathbb{E}_{(x_1,y_1)\sim\mathcal{D}}[\bar{h}(x_1, y_1)]$ for hue shifts. Since, the certified accuracy remains constant with respect to the Wasserstein distance of the shift, we just plot the certified accuracies obtained by various base models trained under different noise levels in Figure 11. We plot the certified accuracies obtained by various models trained using random hue rotations picked uniformly from the range $[-\beta, \beta]$ for different values of the maximum angle $\beta$ in the range. The certified accuracy roughly increases with the training noise achieving a maximum of 87.9% for a max angle $\beta = 180°$ for the training noise level.

## K    RANDOM CHANNEL SELECTION

Consider a smoothing distribution that randomly picks one of the RGB channels with equal probability, scales it so that the maximum pixel value in that channel is one and sets all the other channels to zero. This smoothing distribution is invariant to the color shift transformation $\mathsf{CS}$ and thus, satisfies $\psi(d_{\mathcal{T}}(x_1, x_2)) = 0$ whenever $d_{\mathcal{T}}(x_1, x_2)$ is finite. Therefore, from Theorem 1, we have $\mathbb{E}_{z\sim\tilde{\mathcal{D}}}[\bar{h}(z)] \geq \mathbb{E}_{x\sim\mathcal{D}}[\bar{h}(x)]$ under this smoothing distribution for all Wasserstein bounds $\epsilon$ with respect to $d_{\mathsf{CS}}$. Figure 12 plots the certified accuracies, using random channel selection for smoothing, achieved by models trained using Gaussian distributions of varying noise levels in the transformation space. The certified accuracy roughly increases with the training noise achieving a maximum of 87.1% for a training noise of 0.8.

## L    EXPERIMENTAL DETAILS FOR SECTION 7

Our experimental setting is adapted from the "sample-wise perturbation" CIFAR-10 experiments in Huang et al. (2021): hyperparameters are the same as in that work unless otherwise stated. For background, Huang et al. (2021) creates an unlearnable dataset by performing the following "bi-level" minimization, to simultaneously train a proxy classifier model and create unlearnable examples:

$$\min_{\theta} \min_{(\epsilon_1,\dots,\epsilon_n)} \frac{1}{n} \sum_{i=1}^{n} \mathcal{L}(f_\theta(x_i + \epsilon_i), y_i) \tag{13}$$

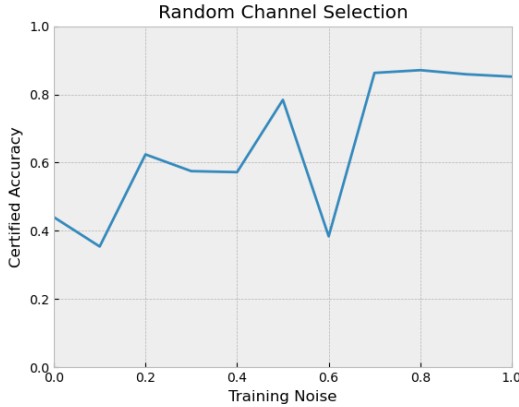

Figure 12: Certified robustness against color shift using random channel selection as the smoothing distribution. Since, the certified accuracy remains constant with respect to the Wasserstein distance ($\epsilon$) of the shifted distribution, we plot the certified accuracy of models trained with various levels of Gaussian noise in the transformation space.

In other words, in contrast with standard training, both the samples and the proxy classifier are optimized to decrease the loss. New classifiers trained on the resulting samples fail to generalize to unperturbed samples. In the experiments, as in Huang et al. (2021), the inner minimization over perturbations is performed for 20 steps over the entire dataset, for every one batch update step of the outer minimization. Training stops when training accuracy reaches a threshold value of 99%.

We now detail differences in experimental setup from Huang et al. (2021):

### L.1 ADAPTATION TO $\ell_2$ ATTACK SETTING

After each optimization step, we project $\epsilon_i$'s into an $\ell_2$ ball (of radius given by the Wasserstein bound $\epsilon$) rather than an $\ell_\infty$ ball. We also use an $\ell_2$ PGD step:

$$\epsilon_i' = \epsilon_i + \tau \frac{\nabla_{\epsilon_i} \mathcal{L}(\cdot)}{\|\nabla_{\epsilon_i} \mathcal{L}(\cdot)\|_2} \tag{14}$$

Step size $\tau$ was set as 0.1 times the total $\ell_2$ $\epsilon$ bound.

### L.2 ADAPTATION TO OFFLINE SETTING

As discussed in the test, we modify the algorithm such that the simultaneous training of the proxy model and generation of perturbations does not introduce statistical dependencies between perturbed training samples. This is especially important because, if the victim later makes a train-validation split, this would introduce statistical dependencies between training and validation samples, making it hard to generalize certificates to a test set.

To avoid this, we construct four data splits:

- Test set (10000 samples): The original CIFAR-10 test set. Never perturbed, only used in final model evaluation.
- Proxy training set (20000 samples): Used for the optimization of the proxy classifier model parameters $\theta$ in Equation 13 and discarded afterward.
- Training set (20000 samples): Perturbed using one round of the the standard 20 steps of the inner optimization of Equation 13, while keeping $\theta$ fixed.
- Validation set (10000 samples): Perturbed using the same method as the "Training set."

The victim model is trained on the "Training Set" and evaluated on the "Validation set" and "Test set". We also tested on the clean (unperturbed) version of the validation set.

## L.3 ADAPTIVE ATTACK SETTING

When testing our smoothing algorithm, we tested two types of attacks:

- Non-adaptive attack: the proxy model is trained and perturbations are generated using undefended models without smoothing: only the victim policy applies smoothing noise during training and evaluation.
- Adaptive attack: In the minimization of Equation 13, the loss term $\mathcal{L}(f_\theta(x_i + \epsilon_i), y_i)$ is replaced by the expectation:

$$\mathbb{E}_{\delta \sim \mathcal{N}(0, \sigma^2 I)} \mathcal{L}(f_\theta(x_i + \epsilon_i + \delta), y_i) \tag{15}$$

In other words, this models the expectation of a *smoothed* model, like the victim classifier. This smoothed optimization is used in both the proxy model training, as well as the generation of the training and validation sets. Following Salman et al. (2019), which proposed a similar adaptive attack for sample-wise smoothed classifiers we approximate the expectation using a small number of random perturbations, which are held fixed for the 20 steps of the inner optimization. In our experiments, we use 8 samples for approximation. Because, at large smoothing noises, this makes the attack much less effective, we cut off training after 20 steps of the outer maximization, rather than relying on the accuracy to reach 99%. (the maximum number of steps required to converge we observed for the non-adaptive attack was 15).

## L.4 RESULTS

Complete experimental results are presented in Figure 13. All results are means of 5 independent runs, and error bars represent standard errors of the means.

# M CONCLUSION

We show that it is possible to certify the distributional robustness of a general deep neural network without increasing its computational requirements. We obtain robustness guarantees with respect to the Wasserstein distance of the distribution shift which is a more suitable metric for out-of-distribution shifts than divergence measures such as KL-divergence and total variation. We only consider predefined distance functions in this work which may not be suitable for capturing more sophisticated distribution shifts such as perceptual changes. A future direction of research could be to adapt our certificates for learnable transformations for domain generalization and adaptation.

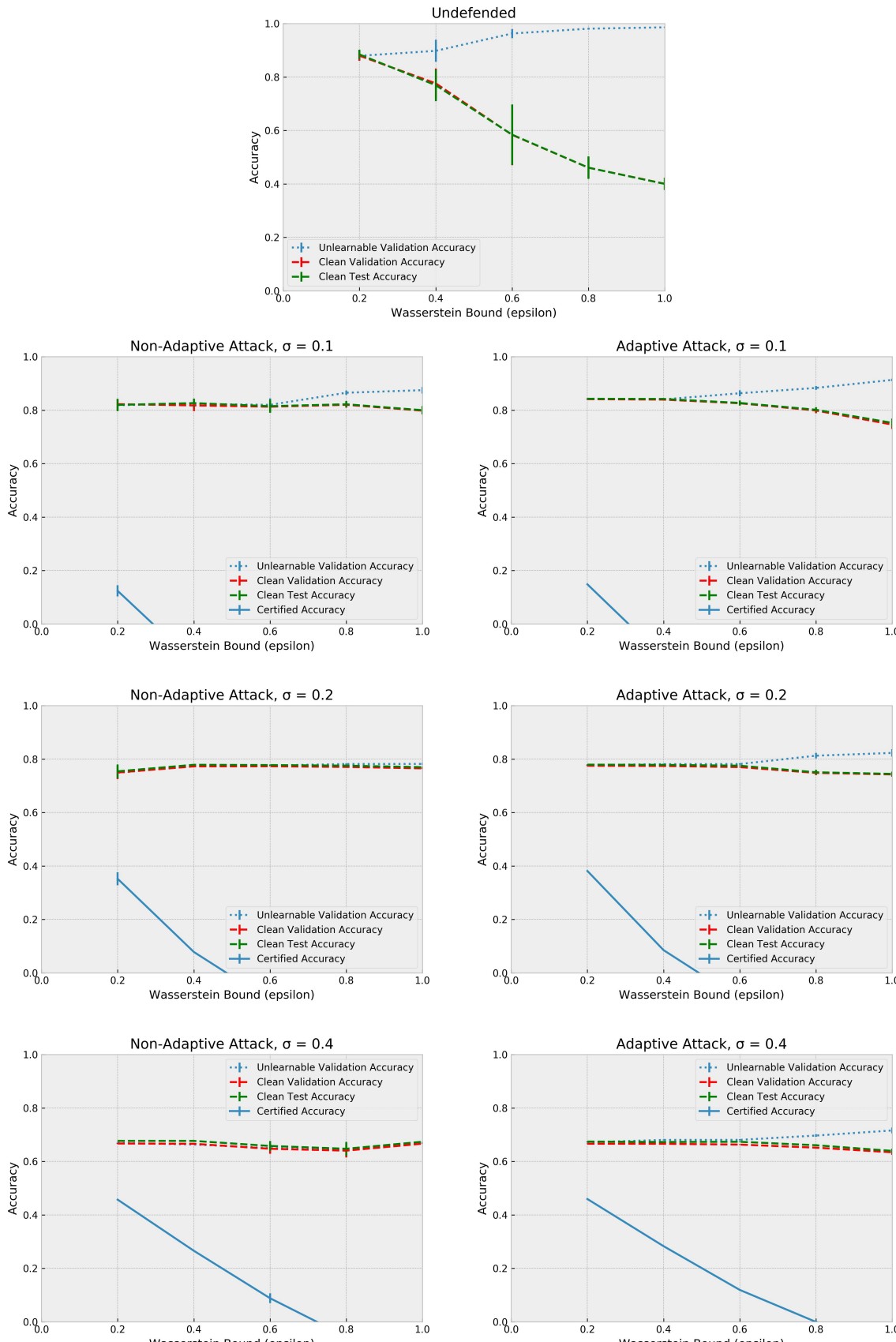

Figure 13: Complete Experimental results for unlearnability experiments.

