# OpenReview forum: "Provable Robustness against Wasserstein Distribution Shifts via Input Randomization"
_ICLR.cc/2023/Conference — ICLR 2023 poster_

### Official Review · Reviewer_fyNV · 2022-10-21

**Confidence:** 3
**Correctness:** 3
**Technical Novelty And Significance:** 2
**Empirical Novelty And Significance:** 2
**Recommendation:** 6

**Clarity, Quality, Novelty And Reproducibility:**

Overall, the writing quality and clarity is adequate. Please find some comments on quality and clarity in the above section. The work is built on past work within the robustness literature, with limited novelty in proof techniques.

**Strength And Weaknesses:**

Overall, the paper is adequately clear.

Some comments on strengths are:
1. The paper discusses the importance of robustness in machine learning with many examples and references to literature.
2. The problem in question itself is an important topic and relevant to ICLR. The paper further contributes to the field by providing interesting insights and results.

Some comments on weaknesses:
1. Why is a bound between the difference of the expectations of smoothed versions meaningful? From a practical view, would it be more reasonable and interesting to have a result that somehow bounds a smoothed expectation with the original, unsmoothed expectation?
2. Taking a brief look at the proof of Theorem 1, it appears that a concave increasing psi is required as an artifact of the proof; is there any additional insight/comments on other reasons why a concave increasing psi is necessary for the definition of (3) and its impact on Theorem 1?
3. More insights can be added to the experiments. In particular, the paper provides calculations of certified accuracy for various examples, but comparisons are lacking. It is difficult to see if the "certifications" are tight. The practicality of these certifications also can use some more explanation.
4. It would be helpful to include discussions on novelties in proof techniques, if any, within the main text.

**Summary Of The Paper:**

From a high level viewpoint, the main result of this paper is a robustness guarantees on models under "smoothing": the paper makes the claim that by smoothing inputs, the difference between smoothed expectations under D and D' with a Wasserstein distance within epsilon is bounded by some function psi of epsilon satisfying a TV-distance "lipshitz type bound" for the smoothing operation. Some applications and discussions on color shifts, hue shifts, brightness shifts, etc., are provided in the paper.

**Summary Of The Review:**

Overall, I think the paper is well written but with several concerns as listed above. I would be happy to adjust my score after discussions with other reviewers and the authors.

---

> ### Author Response · Authors · 2022-11-18
> **Authors' Response**
>
> Thank you for your detailed feedback to help improve our work. We have included your suggestions in the revised version of our manuscript. The following is our response to the concerns raised in the review:
>
> 1. **"Why is a bound between the difference of the expectations of smoothed versions meaningful? From a practical view, would it be more reasonable and interesting to have a result that somehow bounds a smoothed expectation with the original, unsmoothed expectation?"** &mdash; Randomized smoothing works by "averaging" a classifier's predictions on multiple points around the original input point.
>     The robustness of the smoothed classifier comes from the fact that the model is correct on most of these points.
>     The undefended/unsmoothed model evaluates only on one input point which may not be representative of the majority of points around it.
>     While ideally, we would like to have a robustness guarantee on the original model, due to the aforementioned difficulty it is common  to derive robustness bounds on the smoothed model instead.
>     Prior works on randomized smoothing-based approaches also bound the performance of the smoothed model instead of the unsmoothed one. Following are a few examples:
>     - Certified adversarial robustness via randomized
> smoothing, Cohen et. al., ICML 2019.
>     - Certified robustness to adversarial examples with differential privacy, Lecuyer et. al., Symposium on Security
> and Privacy, SP 2019.
>     - Provably robust deep learning via adversarially trained smoothed classifiers, Salman et. al., NeurIPS 2019.
>
> 2. **"Taking a brief look at the proof of Theorem 1, it appears that a concave increasing psi is required as an artifact of the proof; is there any additional insight/comments on other reasons why a concave increasing psi is necessary for the definition of (3) and its impact on Theorem 1?"** &mdash; To develop an intuition for the proof, consider a very simple data distribution that consists only of two equally likely input points that are sufficiently far apart from each other.
>     Say that the shifted distribution perturbs the first point by $\epsilon_1$ and the second by $\epsilon_2$.
>     The Wasserstein distance for this shift is equal to $\frac{\epsilon_1 + \epsilon_2}{2}$.
>     The change in accuracy of the smoothed model due to this shift can be bounded by the average of the total variation of the smoothing distributions at the two points, i.e., $\frac{\psi(\epsilon_1) + \psi(\epsilon_2)}{2}$, which can, in turn, be bounded by $\psi(\frac{\epsilon_1 + \epsilon_2}{2})$ only if $\psi$ is convex.
>
>     Note that the convexity of $\psi$ is not a restrictive assumption for our certificate as there always exists a convex function $\psi$ for any smoothing distribution that upper bounds the total variation.
>     This is because the total variation goes from zero to one as the distance between the input point and the perturbed point goes from zero to infinity.
>     One can always obtain a convex function $\psi$ that upper bounds the TV curve by taking a convex hull of the area under this curve.
>     However, finding a $\psi$ that is easy to compute and is a tight upper bound could be a challenging problem depending on the smoothing distribution.
>
> 3. **"More insights can be added to the experiments. In particular, the paper provides calculations of certified accuracy for various examples, but comparisons are lacking. It is difficult to see if the "certifications" are tight. The practicality of these certifications also can use some more explanation."** &mdash; We perform extensive evaluations and comparisons of our certified guarantees (for different training and smoothing noise levels) and empirical performance under several distribution shifts.
>     We present the empirical performance (indicated by dashed lines) for all the distribution shifts we considered in Figures 2, 5, and 6 - 9.
>     These plots show that the certified accuracies are significantly above the empirical accuracy of an undefended model showing that our certified bounds are non-vacuous and meaningful.
>
>     Practicality: Our approach does not increase the computational requirement of the base model (see Page 2, paragraph 4).
>     It does not require multiple noised evaluations of the model during inference like existing randomized smoothing-based approaches, thereby significantly reducing the sample complexity ($\sim 10^6$ times) during inference.
>     To the best of our knowledge, we present the first certified robustness result for distribution shifts under real-world image transformations such as color shifts.

---

> > ### Author Response · Authors · 2022-11-18
> > **Authors' Response contd.**
> >
> > 4. **"It would be helpful to include discussions on novelties in proof techniques, if any, within the main text."** &mdash; Thank you for the constructive suggestion.
> >     The key observation in our theoretical result is that the total variation of the individual smoothing distributions can be upper bounded by a convex function $\psi$ and this upper bound can then be generalized over the entire distribution using Jensen's inequality.
> >     If the overlap between the smoothing distributions between two individual points does not decrease rapidly with the distance between them, then the overlap between the original and shifted distributions augmented with the smoothing distribution is also high.
> >     We have included this discussion in Section 4 of our paper.

---

> > > ### Comment · Reviewer_fyNV · 2022-11-24
> > > **Response to rebuttal**
> > >
> > > Thanks for the detailed rebuttal and responding to my questions. I also agree with other reviewers that adding experiments on other datasets would help to convey the main results better. Overall, I think this is an interesting work with potential and I am happy to revise my recommendation to 6.

---

> > > > ### Author Response · Authors · 2022-11-26
> > > > **Thank you!**
> > > >
> > > > Thank you for the positive response and the score increase. We really appreciate the valuable insights and expertise you contributed towards reviewing our work.

---

### Official Review · Reviewer_ait3 · 2022-10-25

**Confidence:** 3
**Correctness:** 3
**Technical Novelty And Significance:** 3
**Empirical Novelty And Significance:** 3
**Recommendation:** 6

**Clarity, Quality, Novelty And Reproducibility:**

​The major novelty comes from the no free lunch theory. We can say that, there is no model that can work well in all conditions. Most existing methods may work well on some specific dataset with specific distribution shifts. However, there is only a few understanding on what exactly the distribution shift is and how can the model generalize on the shifted distribution. From the no free lunch theory, we can say that the OOD method can work well on some distribution shifts, but not for all of them. To give theoretical proof on this, what ood the model can generalize is definitely a good direction for the development of trustworthy AI and how to develop a good OOD method.

**Strength And Weaknesses:**

Strength:
- The paper extends the certified robustness into the certified distribution shift robustness.
- The paper proposes a very novel new scenario for the guarantee on the natural distribution shift.
- The method is simple and the theory is solid which can even work on the unlearnable scenario.
​
Weakness:
- The paper majorly focuses on how to extend the certified robustness into natural transformation. ​However, more discussions about what properties of models can impact the OOD robustness is also appreciated.
- The experiment on the OOD data can be more comprehensive. Typically, many results are all drawn with the theoretical proof. The empirical analysis is not sufficient. With only CIFAR10 and SVHN datasets, it is not convincing enough to say whether those problems are a practical OOD problem. We can not say that the model trained with a larger dataset, imagenet, really suffers from this kind of natural shift problem. It is possible that the empirical performance can always have higher performance that the theoretical lower bound. In that way, the provable robustness may not be so important.


**Summary Of The Paper:**

This paper extends the recent certified robustness theory with a simple procedure that randomizes the input of the model within a transformation space. It abandons the  $l_p$ distance change assumption, and does not need any restrictive assumptions or global Lipshitz. With the above theoretical analysis, this paper can provide performance guaranteed on the lower bound on the performance with distribution shift (Both OOD problem and adversarial attack). Experiments give specific guarantees for three natural shifts. Further experiments on the poisoned dataset also show empirical success.

**Summary Of The Review:**

The work quality is good. In terms of methodology, this paper is an extension from the robustness domain. The theoretical results are interesting. Some experiments should be extended correspondingly.

---

> ### Author Response · Authors · 2022-11-18
> **Authors' Response**
>
> Thank you for your positive feedback and for highlighting the novelty of our work. Your constructive suggestions have greatly helped improve our paper. The following is our response to the concerns raised in the review:
>
> 1. **"The paper majorly focuses on how to extend the certified robustness into natural transformation. However, more discussions about what properties of models can impact the OOD robustness is also appreciated."** &mdash; We discuss previous works that have studied ways to improve the empirical robustness of a model to OOD shifts in Section 1: Introduction and Section 2: Related Work.
>     These include approaches such as diversifying datasets (Taori et al., 2020), training with natural corruptions
>     (Hendrycks \& Dietterich, 2019), data augmentations (Yang et al., 2019), contrastive learning (Kim
>     et al., 2020; Radford et al., 2021; Ge et al., 2021) and adversarial training (Goodfellow et al., 2015;
>     Madry et al., 2018; Tramer \& Boneh, 2019; Shafahi et al., 2019; Maini et al., 2020).
>     The approach that we use in this work to improve the empirical robustness of our base models is to augment the training data with transformations such as color shifts, hue shifts, Gaussian perturbation, etc. If there is any empirical robustness approach that we have missed in our paper, we would really like to know about it and add it to our discussion.
>
>     The primary goal of our work is to design a certified robustness technique for Wasserstein distribution shifts without relying on any assumptions about the inner workings of the prediction model. Our work is orthogonal to the study of empirical robustness.
>     Thus, *any* approach that aims to improve the empirical robustness of a given model can be coupled with our approach to obtain improved certified guarantees.
>
> 2. **"The experiment on the OOD data can be more comprehensive. Typically, many results are all drawn with the theoretical proof. The empirical analysis is not sufficient."** &mdash;  We present the empirical performance (indicated by dashed lines) for all the distribution shifts we considered in Figures 2, 5, and 6 - 9.
>    We compare the empirical accuracy of the base models under the respective distribution shifts with the certified accuracy obtained using our theoretical guarantee to show that our approach can achieve meaningful and non-vacuous performance guarantees.
> In Figure 8 and 9, we plot the empirical performance of the smoothed models and show that it always remains above the certified bound, providing experimental evidence of the correctness of the bounds.
>     In Figure 1, we also show good certified accuracy for visible shifts in the image distribution.
>
> 3. **"With only CIFAR10 and SVHN datasets, it is not convincing enough to say whether those problems are a practical OOD problem. We can not say that the model trained with a larger dataset, imagenet, really suffers from this kind of natural shift problem. It is possible that the empirical performance can always have higher performance that the theoretical lower bound. In that way, the provable robustness may not be so important."** &mdash; Models trained on ImageNet have also been shown to be vulnerable to natural, non-adversarial input distribution shifts. Following are a few examples from our literature survey in the Introduction section:
>     - Understanding How Image Quality Affects Deep Neural Networks, Samuel Dodge and Lina Karam, 2016.
>     - Generalisation in humans and deep neural networks, Geirhos et. al., NeurIPS 2018
>     - Why do deep convolutional networks generalize so poorly to small image transformations?, Azulay and Weiss, 2019.
>     - Exploring the Landscape of Spatial Robustness, Engstrom et. al., ICML 2019.
>     - Strike (with) a Pose: Neural Networks Are Easily Fooled by Strange Poses of Familiar Objects, Alcorn et. al., CVPR 2019.
>
>     We pick datasets like CIFAR-10 and SVHN due to their manageable size and ease of training while being sufficiently complex for classification tasks. Our goal is to show that our certificates work for a wide range of applications and image transformations.
>     Training models on a large dataset like ImageNet with the computational resources available to us would have significantly restricted the scope of our experimental evaluations as they require training several models with varying levels of input noise.

---

### Official Review · Reviewer_Ne1j · 2022-10-25

**Confidence:** 3
**Correctness:** 4
**Technical Novelty And Significance:** 3
**Empirical Novelty And Significance:** Not applicable
**Recommendation:** 6

**Clarity, Quality, Novelty And Reproducibility:**

The work is somewhat novel and reproducible.
The paper is clear, but some of the important figures (that should be in the paper but are in the supplementary) are unclear


**Strength And Weaknesses:**

Strength:
- The paper is well-written and mostly easy to follow
- The problem of provable-robustness guarantees is important, especially for DNN.
- Giving a bound using Wasserstein distance can be stronger as it handles shifts that change the support

Weaknesses:
-  The work, more or less, reduces the Wasserstein distance to total variation (whose robustness was already explored previously by Ben-David et al) by smoothing. One issue is that smoothing can reduce performance, thus the robustness is on the smoothed classifier and not on the original classifier.
- The paper talks about adversarial attacks but does not show these results in the main paper. I think these results are much more important than the results presented in the paper on color etc. and must be in the main paper.
- All results should be compared to the empirical robustness (although maybe with fewer plots at once as it is too cluttered to understand).
- I found sec. 6 trivial and misleading. If a classifier is guaranteed to be robust, then of course trying to fool it will fail. It is also misleading as hardness to learn normally is distributions over $(x,y)$ that learning algorithms fail to learn from which isn't the case.

**Summary Of The Paper:**

The work proposes to certify for a network the robustness to change in the input distribution for bounded Wasserstein shift.

**Summary Of The Review:**

The idea isn't groundbreaking but important and worthy of publication. Unfortunately, the authors use most of the paper on less important parts of the work and skip the important parts.

---

> ### Author Response · Authors · 2022-11-18
> **Authors' Response**
>
> Thank you for your valuable comments and feedback to help us improve our work. We have incorporated your suggestions in the revised version of our manuscript. Below we address your concerns one by one:
>
> 1. **"One issue is that smoothing can reduce performance, thus the robustness is on the smoothed classifier and not on the original classifier."** &mdash; Randomized smoothing works by "averaging" a classifier's predictions on multiple points around the original input point. The undefended/unsmoothed model evaluates only on one input point which may not be representative of the majority of points around it. This makes it difficult to connect the expected accuracy of the original (undefended) model with that of the smoothed model. While ideally, we would like to have a robustness guarantee on the original model, due to the aforementioned difficulty it is common to derive performance bounds on the smoothed model instead. Prior works on randomized smoothing-based approaches also bound the performance of the smoothed model instead of the original classifier. Following are a few examples:
>     - Certified adversarial robustness via randomized smoothing, Cohen et. al., ICML 2019.
>     - Certified robustness to adversarial examples with differential privacy, Lecuyer et. al., Symposium on Security and Privacy, SP 2019.
>     - Provably robust deep learning via adversarially trained smoothed classifiers, Salman et. al., NeurIPS 2019.
>
>     Adding smoothing noise to the input indeed decreases the performance of the model.
>     However, this can be mitigated by training the base model with input noise.
>     For example, in Figures 3 and 4, we show that as we increase the training noise, the certified performance of the robust model increases significantly even for large smoothing noise (accuracy of almost 90%).
>     Also, in figure 1, we show that our method can achieve good certified accuracy for visible shifts in the input image distribution.
>
> 2. **"The paper talks about adversarial attacks but does not show these results in the main paper. I think these results are much more important than the results presented in the paper on color etc. and must be in the main paper."** &mdash; We have added the adversarial shift results to the main paper in Section 6 (with a longer version still available in the appendix).
>
> 3. **"All results should be compared to the empirical robustness (although maybe with fewer plots at once as it is too cluttered to understand)."** &mdash; We present the empirical performance (indicated by dashed lines) for all the distribution shifts we considered, see Figures 2, 5, and 6 - 9.
>     These plots show that the certified accuracies are significantly above the empirical accuracy of an undefended model showing that our certified bounds are non-vacuous.
>     Figure 2 compares the empirical accuracy, with (dashed yellow lines) and without (dashed black lines) training noise, with the certified accuracy for all three natural shifts.
>     Figure 5 does this comparison for unlearnable data distributions.
>     Figures 6 through 9 compare the empirical accuracy of an undefended model under adversarial attack (dashed black lines) against certified accuracy for different smoothing noise levels.
>     We also plot the empirical performance of the smoothed models in Figures 7 and 8 (colored dashed lines) and show that it always remains above our derived certified bound as experimental evidence of our theoretical result.
>
> 4. **"I found sec. 6 trivial and misleading. If a classifier is guaranteed to be robust, then of course trying to fool it will fail. It is also misleading as hardness to learn normally is distributions over (x, y) that learning algorithms fail to learn from which isn't the case."** &mdash; Thank you for your valuable feedback.
>     It is a bit unclear to us what is meant in these comments.
>     We try to address the concerns raised to the best of our ability. Hope this helps.
>
>     The objective of unlearnability research is to perturb a data distribution in such a way that a classifier trained on this perturbed distribution achieves a high training accuracy but a low test accuracy on the original distribution, e.g., *Unlearnable examples: Making personal data unexploitable, Huang et. al., ICLR, 2021.*
>     Our result shows that a smoothed model that achieves a high accuracy on the perturbed distribution is guaranteed to achieve a high accuracy on the original distribution, thereby showing a negative result for unlearnability.
>     We show that producing unlearnable data distributions is provably hard.

---

> > ### Comment · Reviewer_Ne1j · 2022-11-19
> > **Thank you**
> >
> > Thank you for addressing my concerns, I raised to score to 6.

---

> > > ### Author Response · Authors · 2022-11-19
> > > **Thank you for your response!**
> > >
> > > Thank you for the score increase. We are glad we were able to address your concerns.

---

### Decision · Program_Chairs · 2023-01-20

**Decision:**

Accept: poster

**Justification For Why Not Higher Score:**

The authors propose an interesting novel certification technique. However, the experiments in the paper do not clearly demonstrate the value of this technique for realistic distribution shifts.

**Justification For Why Not Lower Score:**

The authors develop a novel certification technique for certifying robustness under distribution shifts bounded under the Wasserstein distance metric.

**Metareview: Summary, Strengths And Weaknesses:**

The authors extend the randomized smoothing certified defense to bounding the change in the performance of a model under distribution shifts bounded in the Wasserstein distance metric.

Strengths:
1. Novel certification procedure that applies to distribution shifts not handled by prior work.
2. Experiments demonstrating the efficacy of the developed approach.
3. Applying the technique to obtain fundamental limitations on learning from a poisoned dataset.

Weaknesses:
1. Experiments are only done on easier datasets and the scalability of the technique to more complex datasets or prediction tasks is not studied.

**Note From Pc:**

if the above contains the word "oral" or "spotlight" please see: "oral" presentation means -> notable-top-5% and "spotlight" means -> notable-top-25%. As stated in our emails, we are disassociating presentation type from AC recommendations

**Summary Of Ac-Reviewer Meeting:**

No reviewer meeting.